# Post-process correction improves the accuracy of satellite PM$_{2.5}$ retrievals

Andrea Porcheddu[1], Ville Kolehmainen[1], Timo Lähivaara[1], and Antti Lipponen[2]

[1]Department of Technical Physics, University of Eastern Finland, Kuopio, Finland
[2]Finnish Meteorological Institute, Atmospheric Research Centre of Eastern Finland, Kuopio, Finland

**Correspondence:** Andrea Porcheddu (andrea.porcheddu@uef.fi)

**Abstract.** Estimates of PM$_{2.5}$ levels are crucial for monitoring air quality and studying the epidemiological impact of air quality on the population. Currently, the most precise measurements of PM$_{2.5}$ are obtained from ground stations, resulting in limited spatial coverage. In this study, we consider satellite-based PM$_{2.5}$ retrieval, which involves conversion of high-resolution satellite retrieval of Aerosol Optical Depth (AOD) into high-resolution PM$_{2.5}$ retrieval. To improve the accuracy of the AOD to
PM$_{2.5}$ conversion, we employ the machine learning based post-process correction to correct the AOD-to-PM conversion ratio derived from Modern-Era Retrospective analysis for Research and Applications, Version 2 (MERRA-2) reanalysis model data. The post-process correction approach utilizes a fusion and downscaling of satellite observation and retrieval data, MERRA-2 reanalysis data, various high resolution geographical indicators, meteorological data and ground station observations for learning a predictor for the approximation error in the AOD to PM$_{2.5}$ conversion ratio. The corrected conversion ratio is then
applied to estimate PM$_{2.5}$ levels given the high-resolution satellite AOD retrieval data derived from Sentinel-3 observations. The region of study is central Europe during the year 2019. Our model produces PM$_{2.5}$ estimates with a spatial resolution of 100 meters at satellite overpass times with $R^2 = 0.55$ and RMSE = 6.2 $\mu g/m^3$. The corresponding metrics for monthly averages are $R^2 = 0.72$ and RMSE = 3.7 $\mu g/m^3$. Additionally, we have incorporated an ensemble of neural networks to provide error envelopes for machine learning related uncertainty in the PM$_{2.5}$ estimates. The proposed approach can produce accurate high
resolution PM$_{2.5}$ data that can be very useful for air quality monitoring, emission regulation and epidemiological studies.

## 1 Introduction

Poor air quality is one of the most serious environmental health risks of our time. In September 2021, the World Health Organization (WHO) released Global Air Quality Guidelines, revealing clear evidence of the damage air pollution inflicts on human health at even lower concentrations than previously understood (World Health Organization, 2021). WHO estimates that expo-
sure to air pollution causes 7 million premature deaths every year. A key indicator in monitoring air quality and epidemiological studies is the PM$_{2.5}$ parameter, which is the dry mass concentration of fine particulate matter with an aerodynamic diameter of less than 2.5 micrometers (micrograms of particulate matter per cubic meter of air). Fine particulate matter originates from vehicle emissions, coal burning, and industrial emissions, among many other human and natural sources. Epidemiological studies link long exposures to high PM$_{2.5}$ levels to many severe illnesses, such as stroke and cardiovascular and respiratory

diseases (e.g. Pope and Dockery, 2006; Cohen et al., 2017). On a global scale, the magnitude of the $PM_{2.5}$ exposure-related risk for human health is enormous as more than 90% of the world's population lives in areas with annual mean $PM_{2.5}$ levels exceeding the new WHO 2021 air quality guideline of 5 micrograms per cubic meter (annual average) (Health Effects Institute, 2019).

While the knowledge of the health effects of pollution increases continuously, the epidemiological estimates still have
significant uncertainties due to the lack of accurate global air pollution data (Hammer et al., 2020). Networks of ground-based observation stations produce accurate pointwise observations of $PM_{2.5}$ and certain chemical components such as ozone, sulfur dioxide and nitrogen dioxide. These ground station measurements produce relatively accurate data, but the networks consist of only a few thousand irregularly located observation stations, mainly in developed countries, leading to the insufficient spatial coverage of the $PM_{2.5}$ data. To better monitor and understand air quality and pollution sources near real-time global
observations of air quality are needed. The only way to get spatially resolved air quality data is to utilize satellite retrievals.

Satellite retrievals of $PM_{2.5}$ are often based on satellite AOD retrievals and an AOD-to-PM conversion ratio (Health Effects Institute, 2019; van Donkelaar et al., 2013; Zhang and Kondragunta, 2021; Geng et al., 2015). AOD is a columnar optical quantity, whereas $PM_{2.5}$ is the mass concentration of dry aerosol particles at some single point, typically at the surface level. Many factors affect the AOD-to-PM conversion ratio, including the aerosol vertical extinction profile, aerosol type and size
distribution, and relative humidity. These factors are typically unavailable from a single data source, such as data provided by the instruments onboard a satellite, so a simulation-model-based AOD-to-PM ratio is often used. The simulation-model-based AOD-to-PM conversion ratio is typically computed based on meteorology, chemical transport models (CTM) and auxiliary satellite data such as lidar-based aerosol vertical profiles. The $PM_{2.5}$ retrieval at a given location and time is then calculated as a product of the retrieved satellite AOD and the AOD to $PM_{2.5}$ ratio. The current state-of-the-art $PM_{2.5}$ retrieval algorithm
also contains a post-processing step where the retrieved spatial $PM_{2.5}$ estimate is fitted to the ground-based $PM_{2.5}$ station data by a linear geographically weighted regression (van Donkelaar et al., 2016).

Many previous studies use machine learning techniques to convert AOD to $PM_{2.5}$ levels. In particular, (Ibrahim et al., 2022) used a variant of Random Forest called Extremely Randomised Trees (ET) to estimate $PM_{2.5}$ across Europe. (Stafoggia et al., 2019; Schneider et al., 2020) used Random Forest regressors in a multi-stage approach to estimate $PM_{2.5}$ at ground stations
when only $PM_{10}$ measurements were available, to impute AOD values when not accessible and to finally predict $PM_{2.5}$ values across Italy and Great Britain. (Handschuh et al., 2023) considered multiple Random Forest models to evaluate $PM_{2.5}$ levels across Germany using 4 different AOD datasets.

In this paper, we propose a novel approach for high-resolution satellite-based retrieval of $PM_{2.5}$. While the previous studies use machine learning to learn the AOD to $PM_{2.5}$ conversion directly, we take a novel approach where we train the model to
predict the approximation error in the geophysical model based conversion ratio. Our approach retrieves $PM_{2.5}$ at a spatial resolution of 100 m. It is based on the machine learning post-process correction approach, which we developed for the correction of approximation errors in satellite retrievals (Lipponen et al., 2021) and employed for high-resolution spectral aerosol optical depth (AOD) retrieval (POPCORN AOD) from SENTINEL-3 SYNERGY data (Lipponen et al., 2022). In our algorithm development work, we take the spectral, high-resolution Sentinel-3 POPCORN AOD (Lipponen et al., 2022) as the starting point.

Our PM$_{2.5}$ retrieval is based on the AOD-to-PM$_{2.5}$ conversion ratio applied to the POPCORN AOD. The AOD-to-PM$_{2.5}$ ratio is estimated by machine learning techniques utilizing a fusion of collocated ground station-based in-situ PM$_{2.5}$ data, MERRA-2 reanalysis model AOD and PM$_{2.5}$ data, spectral AERONET AOD, satellite-observed spectral top-of-atmosphere reflectances, meteorology data and various high-resolution geographical indicators representing, for example, population density and land surface elevation. Utilizing these data, we employ the post-process correction approach to the estimation of the AOD-to-PM$_{2.5}$

ratio (Lipponen et al., 2021, 2022; Taskinen et al., 2022) and then the high-resolution PM$_{2.5}$ retrieval is obtained as the product of the post-process corrected AOD-to-PM$_{2.5}$ ratio and POPCORN AOD. By using an ensemble of neural networks, we can also provide error envelopes for the machine learning related uncertainty in the PM$_{2.5}$ estimates. The approach is tested with Sentinel-3 data from central Europe in 2019.

## 2 Data

We use various input data variables in computing the estimate for the surface PM$_{2.5}$. We use satellite observation data and retrievals, in-situ observations, and reanalysis model data. This section lists all the variables and data sources used in our work.

### 2.1 Sentinel-3 POPCORN AOD

The Sentinel-3 POPCORN AOD product is based on the post-process corrected Sentinel-3 SYNERGY land AOD product. It offers a spatial resolution of 300 meters and is currently accessible for Sentinel-3A and 3B overpasses, covering five regions of

interest for the year 2019: Central Europe, Eastern USA, Western USA, Southern Africa, and India. Two Sentinel-3 satellites currently flying provide revisit times of less than two days for OLCI and less than one day for the SLSTR instrument at equator. Swath width of the OLCI instrument is 1270 km. SLSTR swath width is 1420 km for the nadir view and 750 km for the oblique view.

The post-process correction is based on a feed forward neural network that was trained to predict the bias in Sentinel-3

Synergy AOD. Sentinel-3-AERONET-collocated data was used as the training data for the neural network and the trained neural network was then used for bias correction and superresolution of the Sentinel-3 AOD (land) data. The idea for post-process correction of satellite AOD retrievals was introduced in Lipponen et al. (2021). For the technical details and accuracy metrics of Sentinel-3 SYNERGY land POPCORN AOD, and related openly available code and data, see Lipponen et al. (2022).

In this work, we use POPCORN AODs at 440, 500, 550, 675, and 870 nm, and the Angstrom exponent derived using

AODs at these wavelengths as inputs for the AOD-to-PM$_{2.5}$ ratio model. POPCORN AODs are the data that bring the accurate AERONET AOD information to the AOD-to-PM$_{2.5}$ conversion.

### 2.2 OpenAQ

OpenAQ (https://openaq.org/) is an open database for air quality data. In this work, we use OpenAQ as our data source for surface in-situ PM$_{2.5}$ observations. OpenAQ provides pointwise air quality measurement data for thousands of stations. The

temporal resolution of the data provided varies by station, 1-hour and daily observations are commonly available. See Figure 1 for a map of OpenAQ stations providing hourly data in our region of intrest.

Some OpenAQ stations report 24 hour average $PM_{2.5}$ every hour.

In this work, we used the 24 hour averages given every hour to estimate hourly $PM_{2.5}$. This was done station-by-station using a Tikhonov regularized (with regularization parameter value 0.05) least-squares fit to unfold the time integrated data into hourly estimates.

In practice, the hourly $PM_{2.5}$ estimates were computed using the formula

$$PM_{2.5,1h} = \left(A^T A + \alpha I\right)^{-1} A^T b, \tag{1}$$

where

$$A = \begin{bmatrix} \frac{1}{24} & \frac{1}{24} & \cdots & \frac{1}{24} & 0 & 0 & \cdots & 0 \\ 0 & \frac{1}{24} & \cdots & \frac{1}{24} & \frac{1}{24} & 0 & \cdots & 0 \\ & & & \vdots & & & & \\ 0 & 0 & \cdots & 0 & 0 & 0 & \cdots & \frac{1}{24} \end{bmatrix}, \tag{2}$$

$$b = \begin{bmatrix} PM_{2.5,24h,1} \\ PM_{2.5,24h,2} \\ \vdots \\ PM_{2.5,24h,N} \end{bmatrix}, \tag{3}$$

$$PM_{2.5,1h} = \begin{bmatrix} PM_{2.5,1h,24} \\ PM_{2.5,1h,25} \\ \vdots \\ PM_{2.5,1h,N} \end{bmatrix}, \tag{4}$$

and $\alpha$ is the regularization parameter. $PM_{2.5,1h,N}$ and $PM_{2.5,24h,N}$ denote the 1 hour and 24 hour average $PM_{2.5}$ at timestep $N$, respectively.

## 2.3 MERRA-2

The Modern-Era Retrospective analysis for Research and Applications, Version 2 (MERRA-2) is NASA's reanalysis model (Randles et al., 2017). MERRA-2 provides us meteorological variables, such as wind fields and temperatures. Furthermore, MERRA-2 reanalysis also has the necessary aerosol and air quality information to compute an estimate for the surface $PM_{2.5}$.

MERRA-2 has a spatial resolution of $0.5°$ x $0.625°$. This is roughly 50 km in Central Europe region. The time-varying MERRA-2 variables we use have the temporal resolution of 1 hour and both instantaneous values or time-averaged values are

given depending on the variable and data product. We also use some MERRA-2 constant variables as inputs for our AOD-to-PM$_{2.5}$ model. See the Appendix A for a list of all variables we have used as inputs in our models from the MERRA-2 re-analysis.

In addition to MERRA-2 provided variables, the following variables are derived using the MERRA-2 meteorology and aerosol-related variables and used in our models as inputs:

– **Relative humidity (RH) at the surface. Equation based on the Clausius-Clapeyron equation (see e.g. Michaelides et al., 2019):**

$$RH = 0.263 \cdot PS \cdot QLML / \exp\left((17.67 \cdot (T2M - 273.15))/(T2M - 29.65)\right)$$

– **Wind direction (WD10M) at 10 meters:**

$$WD10M = \arctan\left(-V10M/U10M\right)$$

– **Wind speed (WS10M) at 10 meters:**

$$WS10M = \sqrt{U10M^2 + V10M^2}$$

– **PM$_{2.5}$ at surface:** (Buchard et al. (2016))

$$PM_{2.5} = (1.375 \cdot SO4SMASS + 1.4 \cdot OCSMASS + BCSMASS + DUSMASS25 + SSSMASS25) \cdot 10^9$$

– **AOD-to-PM$_{2.5}$ ratio $\eta$:**

$$\eta = \frac{PM_{2.5}}{TOTEXTTAU}$$

## 2.4    CALIOP aerosol vertical profile climatology

We use the Cloud-Aerosol Lidar and Infrared Pathfinder Satellite Observation (CALIPSO) Lidar Level-3 Tropospheric Aerosol
Profiles, Cloud Free Data, Standard Version 4-20 data product as one of our input data source (NASA, 2022; Winker et al., 2010). This level-3 climatology data product has spatial resolution of 2.5 deg x 2 deg and temporal resolution of 1 month. We use daytime variables and in the case of missing data, we use the nearest value found in the dataset. We use two variables from this dataset: AOD 63 Percent Below and AOD 90 Percent Below. These variables indicate the vertical height below which 63 and 90 percent of AOD is located on average. This gives us information about the vertical distribution of aerosols in the
atmosphere.

## 2.5    Time variables

Information about the time of day and year are given as inputs for the model. Both the yearly and daily fractions from the beginning of the year and day until the end of year and day, respectively, are mapped to a unit circle and the x and y coordinates of the unit circle points are used as inputs for the model. With this approach, we get very similar values for the end and
beginning of the year and day.

## 2.6 High-resolution geographical indicators

### 2.6.1 OpenStreetMap roads

OpenStreetMap is an open map project and it contains map data with high spatial resolution. We use OpenStreetMap roads as a data source for our model inputs. We compute the distance to the nearest street or highway and use this distance as an input. We use a 100 meter resolution grid for the distances. The paths, streets and highways are all classified as 'highways' in OpenStreetMap and we use only the following sub-classes to only accept roads and highways with car traffic and thus potential $PM_{2.5}$ sources (information from (OpenStreetMap, 2023)). See Appendix A for all the OpenStreetMap road types used to compute the distance to the closest road.

### 2.6.2 NASA Black Marble Night Lights

NASA's Black Marble is a night light product based on Visible Infrared Imaging Radiometer Suite (VIIRS) day/night band (DNB) radiances measured at nighttime. DNB is highly sensitive to light and can therefore detect even very low intensity lights on Earth surface at night. Most of the nighttime lights seen on Earth's surface are due to human activities. As human footprint is well seen in the night lights, we use the NASA Black Marble Night Lights as a proxy variable for the population density and use it as one input for our models. We use Night Light data at spatial resolution of 500 meter as our input based on the yearly data product VNP46A4 (Wang et al., 2020).

### 2.6.3 MODIS land cover type

We use MODIS `MCD12Q1` (Sulla-Menashe and Friedl, 2018) land cover type data product to derive input variables that contain distances to the closest International Geosphere Biosphere Programme (IGBP) land cover types (Loveland and Belward, 1997; Belward et al., 1999). The spatial resolution of the MODIS `MCD12Q1` data product is 500 meters. For the list of IGBP land cover types, see Appendix A.

### 2.6.4 Digital Elevation Model

We use the Advanced Spaceborne Thermal Emission and Reflection Radiometer (ASTER) digital elevation model (DEM) to describe the land surface elevation (Fujisada et al., 2011, 2012; NASA/METI/AIST/Japan Spacesystems, and US/Japan ASTER Science Team, 2019). ASTER DEM has a spatial resolution of 1 arcsecond corresponding to about 30 meters.

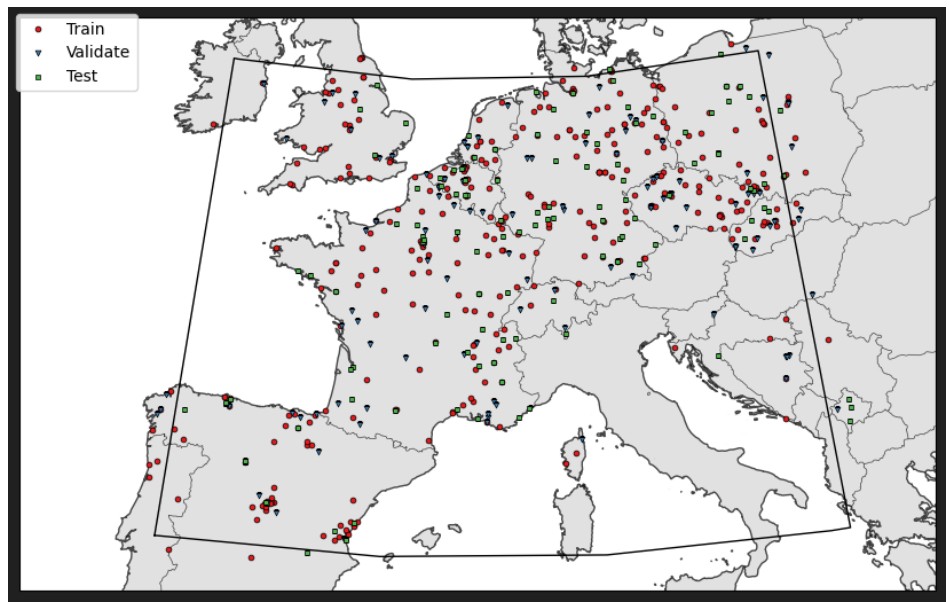

**Figure 1.** Map of stations in the region of interest.

## 3 Methods

### 3.1 AOD-to-PM$_{2.5}$ conversion

Similarly as, for example, in van Donkelaar et al. (2021), we model the dependency between the PM$_{2.5}$ at the surface level and AOD using the following model

$$\mathrm{PM}_{2.5} = \eta \cdot \mathrm{AOD}, \tag{5}$$

where $\eta = \eta(\mathbf{r}, t)$ is the AOD-to-PM$_{2.5}$ conversion coefficient that is function of both time $t$ and space $\mathbf{r}$.

### 3.2 Post-process correction approach

Let $y \in \mathbb{R}^m$ denote an accurate satellite retrieval

$$y = f(x), \tag{6}$$

where vector $y$ contains the output of the satellite retrieval algorithm, $f : \mathbb{R}^n \mapsto \mathbb{R}^m$ is an accurate retrieval algorithm and $x \in \mathbb{R}^n$ contains all the algorithm inputs including the observation geometry and level 1 satellite observation data such as the top-of-atmosphere reflectances. The retrieval $y$ can consist, for example, of surface PM$_{2.5}$ at a given point in space and time.

In practice, due to uncertainties in the auxiliary parameters of the underlying forward model, extensive computational dimension of the problems and processing time limitations, it is not possible to construct an accurate retrieval algorithm $f$ but an

approximate retrieval algorithm

$$\tilde{y} \approx \tilde{f}(x) \tag{7}$$

has to be employed instead. The approximate retrieval $\tilde{f}$ is typically based on physically simplified and computationally reduced approximate forward models that are used due to the huge dimensionality of the retrieval problems and the need for computational efficiency. The utilization of the approximate retrieval algorithm leads to an *approximation error*

$$e(x) = f(x) - \tilde{f}(x) \tag{8}$$

in the retrieval parameters.

The core idea of the *model enforced* post-process correction model is to improve the accuracy of the approximate retrieval (7) by machine learning techniques. By Equations (6)-(8), the accurate retrieval can be written as

$$
\begin{aligned}
y &= f(x) \\
&= \tilde{f}(x) + \left[ f(x) - \tilde{f}(x) \right] \\
&= \tilde{f}(x) + e(x).
\end{aligned}
\tag{9}
$$

To obtain the corrected retrieval, Equation (9) is used to combine the conventional (physics-based) retrieval algorithm $\tilde{f}(x)$ and a machine learning based model $\hat{e}(x)$ to predict the realization of the approximation error $e(x)$ to obtain an corrected retrieval

$$y \approx \tilde{f}(x) + \hat{e}(x). \tag{10}$$

Note that this approach is different from a conventional *fully learned* machine learning model in which the aim is to emulate the accurate retrieval algorithm $f(x)$ with a machine learning model

$$y \approx \hat{f}(x) \tag{11}$$

that is trained to predict the retrieval $y$ directly from the satellite observation and geometry data $x$. The *approximation error* of the physics based retrieval is a less complicated function (compared to the direct retrieval) for a machine learning regression to learn. This leads to a more accurate and reliable estimation of the retrieval quantity.

### 3.3 Correction of AOD-to-PM$_{2.5}$ conversion factor $\eta$

In our work, we use the post-process correction approach (10) to correct for the MERRA-2-based AOD-to-PM$_{2.5}$ conversion factor $\eta$. We utilize an ensemble of neural networks to learn the correction to the conversion factor $\eta$ and producing simultaneously error envelopes related to the learning process. Our post process correction model $\hat{e}(x) : \mathbb{R}^n \mapsto \mathbb{R}$ corrects the conversion factor pixel-by-pixel, meaning that

$$
\begin{aligned}
\eta(x) &= \hat{\eta} + \hat{e}(x) \\
\end{aligned}
\tag{12}
$$
$$
\begin{aligned}
\text{PM}_{2.5} &= \eta(x) \cdot \text{AOD}_{\text{POPCORN}}
\end{aligned}
\tag{13}
$$

where $\hat{\eta}$ represents the AOD-to-PM$_{2.5}$ ratio to be corrected. The correction model is learned using collocated data from ground station PM$_{2.5}$ data, MERRA-2 data, satellite data and retrieval, meteorological data, and high-resolution geographical indicators. All the inputs used can be found in Table A1 and are described in Section 2. We used SHAP analysis (Lundberg and Lee, 2017) in order to estimate feature importance after the training of the model. In fig.A1 you can see a bar plot of the first 26 input features ordered by their importance (SHAP value) and in Table A1 the feature are ordered by their SHAP importance (from left to right and from top to bottom). Since no features showed non-negligible SHAP value, we decided to keep them all in the training of the model. We finally add the estimated correction term to the MERRA-2 $\eta$ values and calculate the PM$_{2.5}$ estimates corresponding to POPCORN AOD retrievals using Equation (5).

## 3.4  Selection of the network model

As the dimension $n$ of the input data $x$ to the correction model $\hat{e}(x)$ is relatively small ($n = 172$) and output is a scalar, we utilize a fully connected feedforward neural network for the regression task. The networks are implemented using the TensorFlow framework.

To optimize the neural network architecture, we employed KerasTuner, a hyperparameter optimization framework. The Adam optimizer and $10^{-3}$ learning rate were selected. We used the Mean Square Error (MSE) loss function in the training. A linear activation function was employed for the output layer as the correction $\hat{e}(x)$ is real valued. Other parameters, such as the activation functions and the number of nodes in hidden layers, were optimized using KerasTuner. We considered the number of hidden layers, experimenting with 2, 3, and 4-layer architectures. The model with two hidden layers led to better accuracy compared to the deeper models with 3 or 4 hidden layers and thus we employed the architecture with two hidden layers as our final model. The final optimal neural network architecture comprises of 172 input features and two hidden layers with seLu activation functions. The first and second hidden layers consisted of 160 and 128 neurons, respectively. Figure 2 shows the neural network architechture obtained from the model optimization.

We divided the dataset into three subsets in training our neural network model. Specifically, 60% of the data was used for training, 20% for validation, and 20% for testing, see Figure 1 for the division of the AQ stations into the training, validation and test sites. The learning data was divided into training, validation and test data by stations instead of random division of data points in order to avoid model overfitting and having test data from locations within the region of interest that were not included in the model training. Figure 3 shows the proportions of different PM$_{2.5}$ values in the train, validate and test data. We used the validation set and the early stopping technique with the patience of 30 to avoid overfitting of the neural network model.

In our tests, the model struggled to predict high PM$_{2.5}$ values accurately. We partially attributed this limitation to the skewed distribution of our dataset, which was predominantly composed of low PM$_{2.5}$ values, see Figure 3 for the histogram of the PM$_{2.5}$ values of the AQ stations in the learning data. To address this, we introduced a cut-off value of 80 $\mu$gm$^{-3}$ for PM$_{2.5}$ and trained our model with samples corresponding to PM$_{2.5}$ values only below this. Furthermore, we experimented with reweighting the loss function to emphasize higher PM$_{2.5}$ values. Although this strategy slightly improved the model's performance on

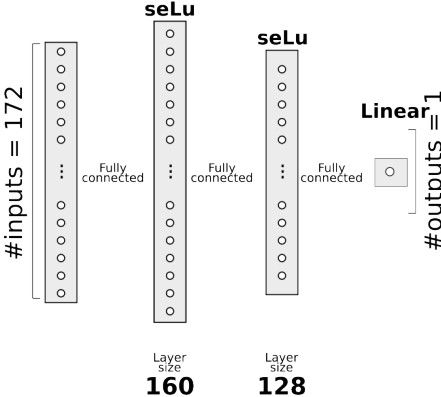

**Figure 2.** Feed-forward neural network architecture for post-process correction of $\eta$ ratio, optimized with KerasTuner. The model contains two hidden layers with seLu activation functions (160 and 128 nodes respectively) and a single node output layer with linear activation function.

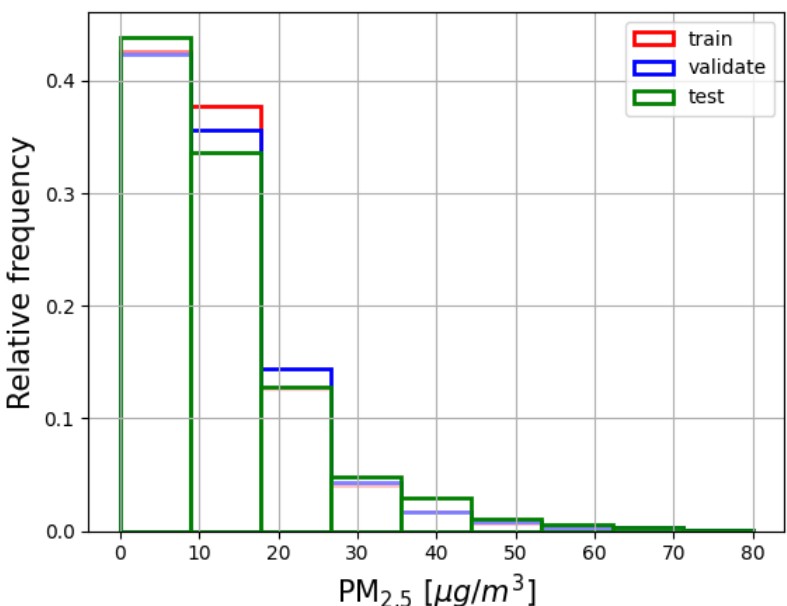

**Figure 3.** Distribution of AQ station $PM_{2.5}$ values in training, validation, and test sets. The training data is used to train the machine learning algorithm, while the validation data is used to prevent overfitting. The test data is used to test the results after training. The division of the data was obtained by dividing the AQ stations in the region of interest to three separate sets with 60%, 20% and 20% shares of training, validation and test stations.

the high-end tail, it compromised the accuracy on the low-end tail. Consequently, we decided not to use the reweighted loss function.

## 3.5 Ensemble of networks

To adress the problem of local minima and dependency on the initialization in neural network training we used an ensemble based technique where we trained an ensemble of 80 networks each initialized with different random weights. We considered
the predictions of the networks as samples from a distribution and used the median of the predictions as a point estimate for the correction term of $\eta$. We use the spread minimum to maximum interval of the 80 outputs of the networks as an learning related uncertainty for $\eta$ which was propagated onward to uncertainty of the $PM_{2.5}$ estimates through the conversion (5).

## 4 Results

Figure 4 shows scatter plots of the satellite and model-based predictions of $PM_{2.5}$ with respect the values of the ground stations
for the test data AQ stations per single-overpass and as monthly averages. We calculated the monthly averages considering a threshold: monthly averages were accepted only when we had more than 5 daily measurements per month (and station). The figures on the top row show results for single-overpasses and the figures on the bottom row show monthly averages. The figures on the left show the ground data comparison for the MERRA-2 $PM_{2.5}$ estimates, the figures on the middle show the ground data comparison for the $PM_{2.5}$ values estimated using Equation (5) with POPCORN AOD and MERRA-2 conversion factor $\eta$,
and the figures on the right show the comparison for the $PM_{2.5}$ values estimated using Equation (5) with POPCORN AOD and post-process corrected $\eta$. As can be seen, the use of post-process corrected conversion factor leads to a clear improvement on the accuracy of the predictions of $PM_{2.5}$ at the independent test data locations. The $R^2$ coefficient for instantaneous values is improved by about $290\%$ compared to both the MERRA-2 prediction and the estimate (5) with POPCORN AOD and MERRA-2 conversion factor. The RMSE is improved by a factor $32\%$ compared to MERRA-2 prediction and by a factor $41\%$ compared
to the product of POPCORN AOD with MERRA-2 $\eta$. The absolute value of the bias is reduced by a factor over $95\%$ respect to both of the uncorrected estimates, and the MAE decreased by a factor $26\%$ compared to MERRA-2 prediction and by a factor $41\%$ compared to the product of POPCORN AOD with MERRA-2 $\eta$. In the monthly averages the $R^2$ coefficient is improved by a factor $350\%$ respect to MERRA-2 prediction and by a factor $279\%$ compared to the estimate (5) with POPCORN AOD and MERRA-2 $\eta$. The RMSE in the monthly averages is reduced by a factor over $47\%$ with respect to both uncorrected methods.
The bias in the monthly averages is reduced by a factor $92\%$ and $89\%$, respectively, and the MAE decreased by a factor $44\%$ and $49\%$.

We remark that we tested also the fully-learned approach (11) for learning directly the AOD-to-$PM_{2.5}$ conversion factor $\eta$ values instead of the correction of the MERRA-2 based conversion, but the results with the fully learned approach were less accurate than with the post-correction approach (10). The comparison can be seen in fig.A2.
Figures 5 and 6 show $PM_{2.5}$ maps over Paris (23 February 2019) and Madrid (29 March 2019) for a single satellite overpass, respectively. On the top-left the uncorrected map is obtained based on POPCORN AOD 500nm and MERRA-2 $\eta$, while on

the top-right the corrected map uses the post-process corrected MERRA-2 $\eta$. On the bottom left we compare the satellite based PM2.5 values to the measured PM2.5 values at the AQ stations which are represented by the circles in the maps. The red circles represent the post-corrected estimates (medians calculated from the ensemble predictions), the black dots the uncorrected estimates while the blue dots the ground based measurement values at the stations. The red error bars represent the spread of $PM_{2.5}$ values coming from the ensemble of networks and they are to be considered as uncertainty estimates related to the machine learning process. The joint RMSE of the uncorrected estimates with respect to the ground stations are 7.82 $\mu g/m^3$ and 4.59 $\mu g/m^3$ respectively for Paris and Madrid, and the joint RMSE for the post-corrected estimates with respect the ground stations are 6.36 $\mu g/m^3$ and 2.27 $\mu g/m^3$, indicating improved accuracy of the per overpass $PM_{2.5}$ estimates in the post process correction approach. The figure reveals that, for all the stations, the different initialization points for the trainings improve over the uncorrected prediction. The median of the ensemble predictions is not always better than the uncorrected prediction, but the uncertainty interval is either enclosing the measured value or is closer to the measured value than the uncorrected estimate. The bottom right images show a time series of $PM_{2.5}$ monthly averages predictions against the time series coming from a ground station monthly averages (the stations are pointed on the corrected maps by a white arrow). The red envelopes show the uncertainty envelope of the post-process corrected estimate. Here the ground station monthly averages are contained in the uncertainty envelope. Figure 7 shows time series of $PM_{2.5}$ monthly averages of the post-process corrected estimates for different stations in the region of interest, showing good alignment with the accurate ground based AQ measurements. Similar performance was found out for the monthly averages in most of the test stations in the region of interest, indicating that the post process corrected estimates of monthly averages of $PM_{2.5}$ are generally well aligned with the accurate ground based observations.

The post process correction method we have proposed here is flexible with respect data to be utilized in the training, as it allows straightforward addition of more training data (by re-optimization of the neural network architecture) coming from different data sources in order to improve the $PM_{2.5}$ predictions. In this study, we demonstrated the approach using POPCORN AOD data, which is obtained post-correcting Sentinel-3 AOD. The approach can also be extended and trained to other satellite instruments and their AOD products to obtain similarly post-process corrected high-resolution satellite estimates of $PM_{2.5}$, leading to more frequent temporal sampling of a particular location. In this study, we demonstrated the approach using a relatively large region-of-interest covering central Europe year 2019. The approach can also be scaled in a straightforward manner to smaller or larger regions of interest by changing the training data. To demonstrate the performance of our approach with different model data, we tested the post-process correction using Goddard Earth Observing System Composition Forecast (GEOS-CF) data (Keller et al., 2021) instead of MERRA-2 data. GEOS-CF offers a higher spatial resolution of 25 km and variables that are not available from MERRA-2, for example additional chemical species such as nitrate. Temporal resolution of GEOS-CF is 1 hour. The result obtained when GEOS-CF data is used in the training of the correction model is shown in fig.8. Comparison to fig. 4 C shows that the performance of the correction model is similar to the model trained with MERRA-2 with MERRA-2 leading to slightly better error metrics.

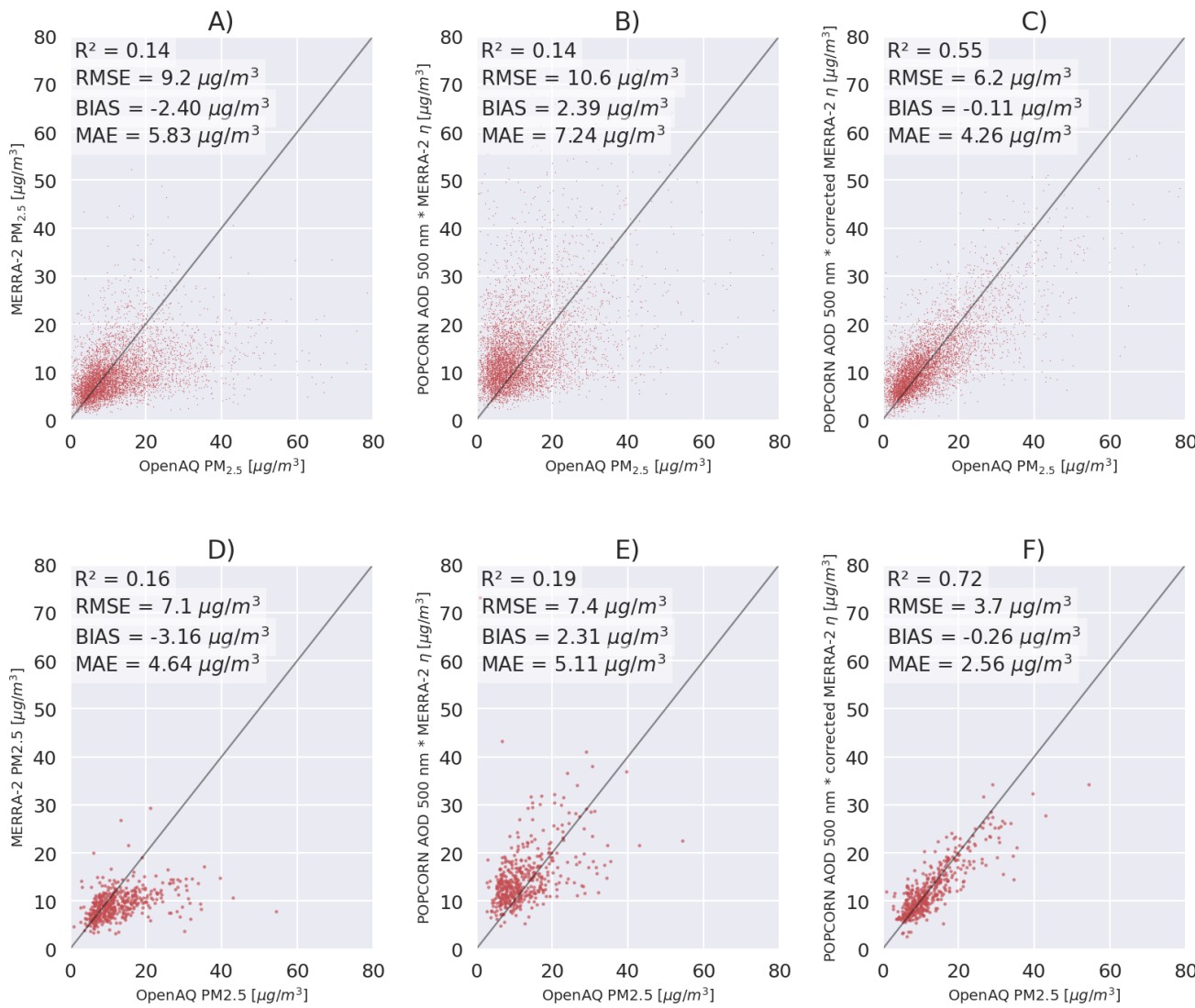

**Figure 4.** A) MERRA-2 $PM_{2.5}$ predictions against OpenAQ $PM_{2.5}$ measurements per single-overpass. B) Uncorrected NOODLESALAD $PM_{2.5}$ predictions against OpenAQ $PM_{2.5}$ measurements per single-overpass. C) Corrected NOODLESALAD $PM_{2.5}$ predictions against OpenAQ $PM_{2.5}$ measurements per single-overpass. D) MERRA-2 monthly averages $PM_{2.5}$ predictions against OpenAQ monthly averages $PM_{2.5}$ measurements. E) Uncorrected NOODLESALAD monthly averages $PM_{2.5}$ predictions against OpenAQ monthly averages $PM_{2.5}$ measurements. F) Corrected NOODLESALAD monthly averages $PM_{2.5}$ predictions against OpenAQ monthly averages $PM_{2.5}$ measurements.

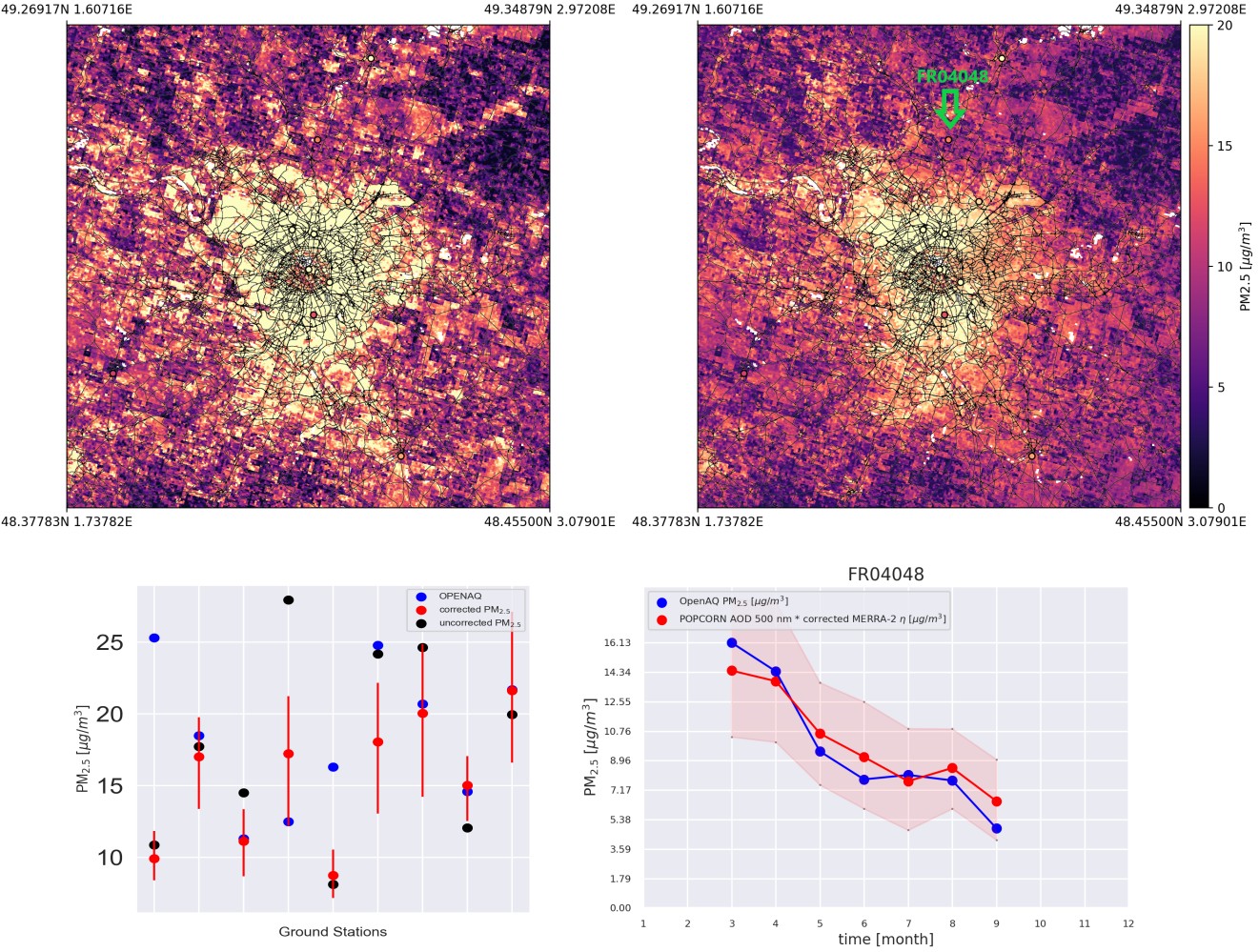

**Figure 5.** On the top-left: single overpass not-corrected PM$_{2.5}$ map over Paris (RMSE against ground stations = 7.82 $\mu g/m^3$). On the top-right: single overpass corrected PM$_{2.5}$ map over Paris (RMSE against ground stations = 6.36 $\mu g/m^3$). Notice that the white regions for the figures on top are regions where the AOD (so the PM$_{2.5}$) values are missing because of cloud contamination. On the bottom-left: comparison of the uncorrected and corrected method at the ground stations, The red error bars represent the spread of values obtained through the ensemble method, while the red dots represent the medians of those values. On the bottom-right: comparison between OpenAQ and corrected method predicted time series of PM$_{2.5}$ monthly averages at a single station (indicated on the corrected map by a green arrow). The red envelope represents the uncertainty coming from the ensemble method.

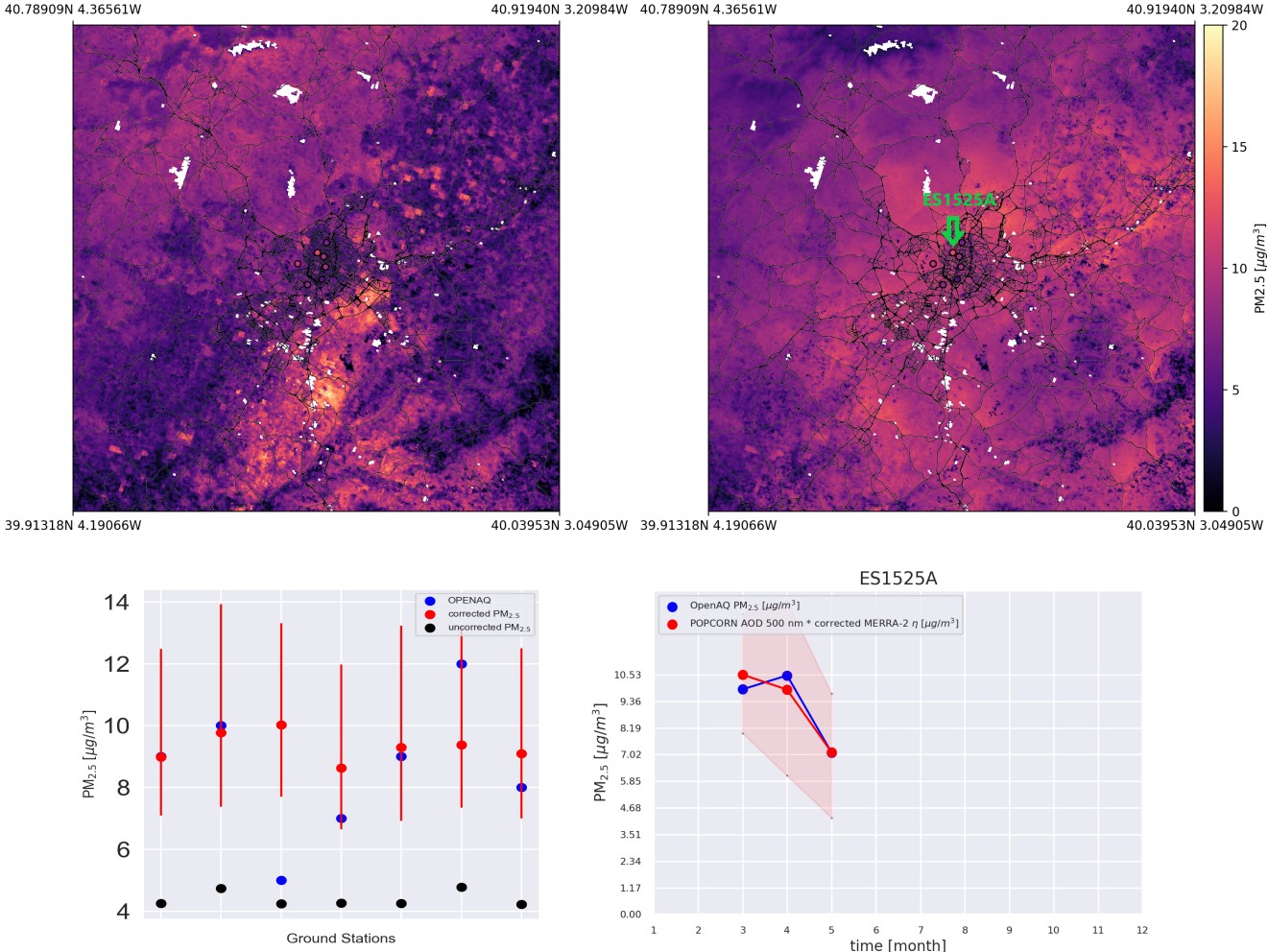

**Figure 6.** On the top-left: single overpass not-corrected PM$_{2.5}$ map over Madrid (RMSE against ground stations = 4.59 $\mu g/m^3$). On the top-right: single overpass corrected PM$_{2.5}$ map over Madrid (RMSE against ground stations = 2.27 $\mu g/m^3$). Notice that the white regions for the figures on top are regions where the AOD (so the PM$_{2.5}$) values are missing because of cloud contamination. On the bottom-left: comparison of the uncorrected and corrected method at the ground stations. The red error bars represent the spread of values obtained through the ensemble method, while the red dots represent the medians of those values. On the bottom-right: comparison between OpenAQ and corrected method predicted time series of PM$_{2.5}$ monthly averages at a single station (indicated on the corrected map by a green arrow). The red envelope represents the uncertainty coming from the ensemble method.

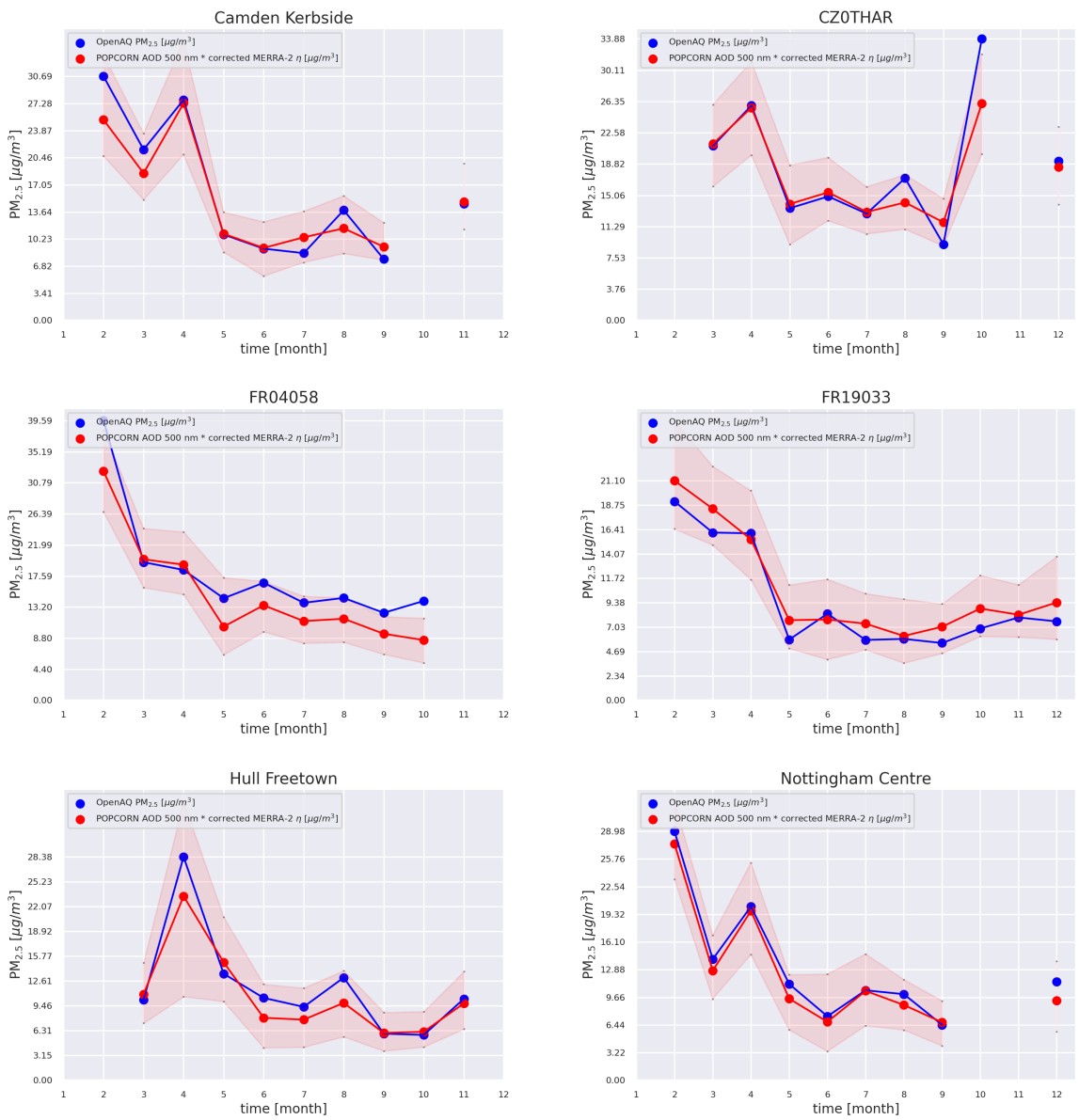

**Figure 7.** Monthly averages time series for six stations from the independent test set within the region of interest. The red envelopes represent the uncertainty coming from the ensemble method.

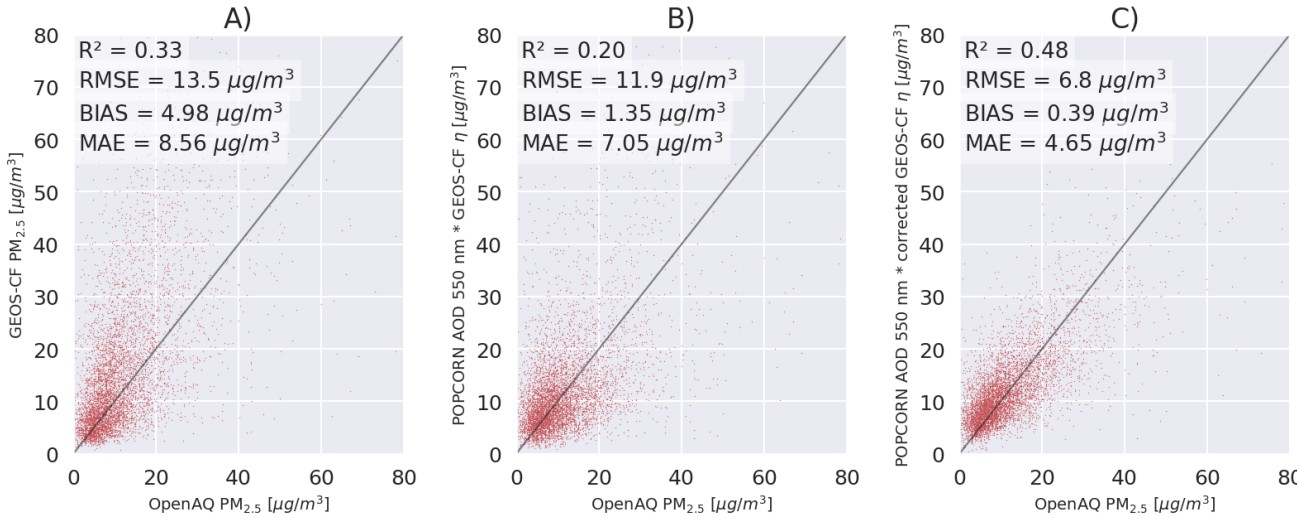

**Figure 8.** A) GEOS-CF PM$_{2.5}$ predictions against OpenAQ PM$_{2.5}$ measurements per single-overpass. B) Uncorrected NOODLESALAD PM$_{2.5}$ predictions against OpenAQ PM$_{2.5}$ measurements per single-overpass (using GEOS-CF data). C) Corrected NOODLESALAD PM$_{2.5}$ predictions against OpenAQ PM$_{2.5}$ measurements per single-overpass (using GEOS-CF data).

## 5    Conclusions

We developed an innovative machine learning technique aimed at correcting the AOD-to-PM$_{2.5}$ ratio derived from MERRA-2 data. This correction method integrates data from various sources, including ground station PM$_{2.5}$ data, MERRA-2 data, satellite data, meteorological data, and high-resolution geographical indicators. The post process corrected AOD-to-PM ratio was then employed to estimate PM$_{2.5}$ levels within the Central Europe region for the year 2019. Our approach outperforms MERRA-2 predictions and predictions made using MERRA-2 AOD-to-PM ratio and POPCORN AOD, resulting in improvement in all evaluated metrics, whether considering individual overpasses or monthly averages. The PM$_{2.5}$ estimates were derived by aggregating the median values from an ensemble of neural networks. We incorporated the ensemble's value spread as a measure of machine learning related uncertainty in the post-process corrected PM$_{2.5}$ estimates, and our estimates with their uncertainty envelopes were found to be generally highly feasible with respect the accurate ground based observations at the independent test station locations. We remark that while our approach produced generally good accuracy in estimation of PM$_{2.5}$, it exhibited poorer performance for the high end values of PM$_{2.5}$. This finding can be attributed to small number of learning data for the high end tail of PM$_{2.5}$ values in our region of interest, highlighting the obvious fact that the learning data for machine learning needs to be representative for the operational environment and conditions.

In this study, our goal was to utilize a simple neural network model to estimate the PM$_{2.5}$ values from satellite data. Therefore, the adoption of a fully connected neural network architecture was considered a reasonable choice. The architecture of the network was determined through a combination of manual selection and the use of KerasTuner to optimize the number of

neurons per layer and the activation function. This ensured the development of an effective network for the specific problem under study. The robust performance of the resulting model highlights the efficacy of employing a simple neural network model to address $PM_{2.5}$ estimation with notable success.

*Code and data availability.* Sentinel-3 Synergy Land POPCORN dataset is openly available for download at https://a3s.fi/swift/v1/AUTH_ca5072b7b22e463b85a2739fd6cd5732/POPCORNdata/readme.html. The OpenAQ data is open data and available for download at https://openaq.org/. The OpenStreetMap data is open data and available for download at https://www.openstreetmap.org/. All the NASA data (MERRA-2, CALIOP, MODIS, ASTER DEM) used in this work is open data and can be found and downloaded using the NASA Earthdata Search website at https://www.earthdata.nasa.gov/. The NASA Black Marble Night Lights data is available at https://blackmarble.gsfc.nasa.

gov/. Code will be available from the authors on a reasonable request.

*Author contributions.* **Andrea Porcheddu**: Conceptualization, Methodology, Software, Formal analysis, Writing — Original draft, Visualization **Ville Kolehmainen**: Conceptualization, Methodology, Formal analysis, Writing — Original draft, Supervision **Timo Lähivaara**: Conceptualization, Methodology, Formal analysis, Writing - Original Draft, Supervision **Antti Lipponen**: Conceptualization, Methodology, Software, Formal analysis, Writing — Original draft, Visualization, Supervision

*Competing interests.* The authors declare no competing interests.

*Acknowledgements.* This study was funded by the European Space Agency EO Science for Society programme via the NOODLESALAD project (contract number 4000137651/22/I-DT-lr). The research was also supported by the Research Council of Finland via the Finnish Centre of Excellence of Inverse Modelling and Imaging (project no. 353084), Flagship of Advanced Mathematics for Sensing Imaging and Modelling (grant no. 358944), and research project (grant no. 321761). The authors wish to acknowledge CSC – IT Center for Science,

Finland, for computational resources.

**Appendix A: Lists of variables used from datasets**

**A1 MERRA-2 variables**

We use the following meteorology-related variables from the MERRA-2 `M2T1NXSLV` dataset:

- **PS**: surface pressure (Pa)
- **QV10M**: 10-meter specific humidity (kg kg$^{-1}$)
- **QV2M**: 2-meter specific humidity (kg kg$^{-1}$)
- **SLP**: sea level pressure (Pa)
- **T10M**: 10-meter air temperature (K)
- **T2M**: 2-meter air temperature (K)

- **TO3**: total column ozone (Dobsons)
- **TOX**: total column odd oxygen (kg m$^{-2}$)
- **TQI**: total precipitable ice water (kg m$^{-2}$)
- **TQL**: total precipitable liquid water (kg m$^{-2}$)
- **TQV**: total precipitable water vapor (kg m$^{-2}$)

- **TROPPB**: tropopause pressure based on blended estimate (Pa)
- **TROPPT**: tropopause pressure based on thermal estimate (Pa)
- **TROPPV**: tropopause pressure based on EPV estimate (Pa)
- **TROPQ**: tropopause specific humidity using blended TROPP estimate (kg kg$^{-1}$)
- **TROPT**: tropopause temperature using blended TROPP estimate (K)

- **TS**: surface skin temperature (K)
- **U10M**: 10-meter eastward wind (m / s)
- **U2M**: 2-meter eastward wind (m / s)
- **U50M**: eastward wind at 50 meters (m / s)
- **V10M**: 10-meter northward wind (m / s)

– **V2M**: 2-meter northward wind (m / s)

    – **V50M**: northward wind at 50 meters (m / s)

We use the following meteorology-related variables from the MERRA-2 `M2T1NXFLX` dataset:

    – **BSTAR**: surface bouyancy scale ($\text{m s}^{-2}$)

    – **CDH**: surface exchange coefficient for heat ($\text{kg m}^{-2}\,\text{s}^{-1}$)

– **CDM**: surface exchange coefficient for momentum ($\text{kg m}^{-2}\,\text{s}^{-1}$)

    – **CDQ**: surface exchange coefficient for moisture ($\text{kg m}^{-2}\,\text{s}^{-1}$)

    – **CN**: surface neutral drag coefficient (1)

    – **DISPH**: zero plane displacement height (m)

    – **EFLUX**: total latent energy flux ($\text{W m}^{-2}$)

– **EVAP**: evaporation from turbulence ($\text{kg m}^{-2}\,\text{s}^{-1}$)

    – **FRCAN**: areal fraction of anvil showers (1)

    – **FRCCN**: areal fraction of convective showers (1)

    – **FRCLS**: areal fraction of nonanvil large scale showers (1)

    – **FRSEAICE**: ice covered fraction of tile (1)

– **GHTSKIN**: ground heating for skin temp ($\text{W m}^{-2}$)

    – **HFLUX**: sensible heat flux from turbulence ($\text{W m}^{-2}$)

    – **HLML**: surface layer height (m)

    – **NIRDF**: surface downwelling nearinfrared diffuse flux ($\text{W m}^{-2}$)

    – **NIRDR**: surface downwelling nearinfrared beam flux ($\text{W m}^{-2}$)

– **PBLH**: planetary boundary layer height (m)

    – **PGENTOT**: total column production of precipitation ($\text{kg m}^{-2}\,\text{s}^{-1}$)

    – **PRECANV**: anvil precipitation ($\text{kg m}^{-2}\,\text{s}^{-1}$)

    – **PRECCON**: convective precipitation ($\text{kg m}^{-2}\,\text{s}^{-1}$)

- **PRECLSC**: nonanvil large scale precipitation (kg m$^{-2}$ s$^{-1}$)

- **PRECSNO**: snowfall (kg m$^{-2}$ s$^{-1}$)

- **PRECTOT**: total precipitation from atm model physics (kg m$^{-2}$ s$^{-1}$)

- **PRECTOTCORR**: Bias corrected total precipitation (kg m$^{-2}$ s$^{-1}$)

- **PREVTOT**: total column re-evap/subl of precipitation (kg m$^{-2}$ s$^{-1}$)

- **QLML**: surface specific humidity (1)

- **QSH**: effective surface specific humidity (kg kg$^{-1}$)

- **QSTAR**: surface moisture scale (kg kg$^{-1}$)

- **RHOA**: air density at surface (kg m$^{-3}$)

- **RISFC**: surface bulk Richardson number (1)

- **SPEED**: surface wind speed (m s$^{-1}$)

- **SPEEDMAX**: surface wind speed (m s$^{-1}$)

- **TAUGWX**: surface eastward gravity wave stress (N m$^{-2}$)

- **TAUGWY**: surface northward gravity wave stress (N m$^{-2}$)

- **TAUX**: eastward surface stress (N m$^{-2}$)

- **TAUY**: northward surface stress (N m$^{-2}$)

- **TCZPBL**: transcom planetary boundary layer height (m)

- **TLML**: surface air temperature (K)

- **TSH**: effective surface skin temperature (K)

- **TSTAR**: surface temperature scale (K)

- **ULML**: surface eastward wind (m s$^{-1}$)

- **USTAR**: surface velocity scale (m s$^{-1}$)

- **VLML**: surface northward wind (m s$^{-1}$)

- **Z0H**: surface roughness for heat (m)

- **Z0M**: surface roughness (m)

We use the following aerosol and air quality related variables from the MERRA-2 `M2T1NXAER` dataset:

- **BCANGSTR**: Black Carbon Angstrom parameter [470-870 nm] (1)

- **BCCMASS**: Black Carbon Column Mass Density (kg m$^{-2}$)

- **BCEXTTAU**: Black Carbon Extinction AOT [550 nm] (1)

- **BCFLUXU**: Black Carbon column u-wind mass flux (kg m$^{-1}$ s$^{-1}$)

- **BCFLUXV**: Black Carbon column v-wind mass flux (kg m$^{-1}$ s$^{-1}$)

- **BCSCATAU**: Black Carbon Scattering AOT [550 nm] (1)

- **BCSMASS**: Black Carbon Surface Mass Concentration (kg m$^{-3}$)

- **DMSCMASS**: DMS Column Mass Density (kg m$^{-2}$)

- **DMSSMASS**: DMS Surface Mass Concentration (kg m$^{-3}$)

- **DUANGSTR**: Dust Angstrom parameter [470-870 nm] (1)

- **DUCMASS**: Dust Column Mass Density (kg m$^{-2}$)

- **DUCMASS25**: Dust Column Mass Density - PM 2.5 (kg m$^{-2}$)

- **DUEXTT25**: Dust Extinction AOT [550 nm] - PM 2.5 (1)

- **DUEXTTAU**: Dust Extinction AOT [550 nm] (1)

- **DUFLUXU**: Dust column u-wind mass flux (kg m$^{-1}$ s$^{-1}$)

- **DUFLUXV**: Dust column v-wind mass flux (kg m$^{-1}$ s$^{-1}$)

- **DUSCAT25**: Dust Scattering AOT [550 nm] - PM 2.5 (1)

- **DUSCATAU**: Dust Scattering AOT [550 nm] (1)

- **DUSMASS**: Dust Surface Mass Concentration (kg m$^{-3}$)

- **DUSMASS25**: Dust Surface Mass Concentration - PM 2.5 (kg m$^{-3}$)

- **OCANGSTR**: Organic Carbon Angstrom parameter [470-870 nm] (1)

- **OCCMASS**: Organic Carbon Column Mass Density (kg m$^{-2}$)

- **OCEXTTAU**: Organic Carbon Extinction AOT [550 nm] (1)

- **OCFLUXU**: Organic Carbon column u-wind mass flux (kg m$^{-1}$ s$^{-1}$)

- **OCFLUXV**: Organic Carbon column v-wind mass flux (kg m$^{-1}$ s$^{-1}$)

- **OCSCATAU**: Organic Carbon Scattering AOT [550 nm] (1)

- **OCSMASS**: Organic Carbon Surface Mass Concentration (kg m$^{-3}$)

- **SO2CMASS**: SO2 Column Mass Density (kg m$^{-2}$)

- **SO2SMASS**: SO2 Surface Mass Concentration (kg m$^{-3}$)

- **SO4CMASS**: SO4 Column Mass Density (kg m$^{-2}$)

- **SO4SMASS**: SO4 Surface Mass Concentration (kg m$^{-3}$)

- **SSANGSTR**: Sea Salt Angstrom parameter [470-870 nm] (1)

- **SSCMASS**: Sea Salt Column Mass Density (kg m$^{-2}$)

- **SSCMASS25**: Sea Salt Column Mass Density - PM 2.5 (kg m$^{-2}$)

- **SSEXTT25**: Sea Salt Extinction AOT [550 nm] - PM 2.5 (1)

- **SSEXTTAU**: Sea Salt Extinction AOT [550 nm] (1)

- **SSFLUXU**: Sea Salt column u-wind mass flux (kg m$^{-1}$ s$^{-1}$)

- **SSFLUXV**: Sea Salt column v-wind mass flux (kg m$^{-1}$ s$^{-1}$)

- **SSSCAT25**: Sea Salt Scattering AOT [550 nm] - PM 2.5 (1)

- **SSSCATAU**: Sea Salt Scattering AOT [550 nm] (1)

- **SSSMASS**: Sea Salt Surface Mass Concentration (kg m$^{-3}$)

- **SSSMASS25**: Sea Salt Surface Mass Concentration - PM 2.5 (kg m$^{-3}$)

- **SUANGSTR**: SO4 Angstrom parameter [470-870 nm] (1)

- **SUEXTTAU**: SO4 Extinction AOT [550 nm] (1)

- **SUFLUXU**: SO4 column u-wind mass flux (kg m$^{-1}$ s$^{-1}$)

- **SUFLUXV**: SO4 column v-wind mass flux (kg m$^{-1}$ s$^{-1}$)

- **SUSCATAU**: SO4 Scattering AOT [550 nm] (1)

- **TOTANGSTR**: Total Aerosol Angstrom parameter [470-870 nm] (1)

- **TOTEXTTAU**: Total Aerosol Extinction AOT [550 nm] (1)

- **TOTSCATAU**: Total Aerosol Scattering AOT [550 nm] (1)

## A2   OpenStreetMap road types used to compute the distance to the closest road

We use the following road types to compute the distance to the closest road. The descriptions of the road types are obtained from OpenStreetMap (2023).

- **motorway**: A restricted access major divided highway, normally with 2 or more running lanes plus emergency hard shoulder. Equivalent to the Freeway, Autobahn, etc.

- **trunk**: The most important roads in a country's system that aren't motorways.

- **primary**: The next most important roads in a country's system.

- **secondary**: The next most important roads in a country's system.

- **tertiary**: The next most important roads in a country's system.

- **motorway_link**: The link roads (sliproads/ramps) leading to/from a motorway from/to a motorway or lower class highway. Normally with the same motorway restrictions.

- **trunk_link**: The link roads (sliproads/ramps) leading to/from a trunk road from/to a trunk road or lower class highway.

- **primary_link**: The link roads (sliproads/ramps) leading to/from a primary road from/to a primary road or lower class highway.

- **secondary_link**: The link roads (sliproads/ramps) leading to/from a secondary road from/to a secondary road or lower class highway.

- **tertiary_link**: The link roads (sliproads/ramps) leading to/from a tertiary road from/to a tertiary road or lower class highway.

## A3   IGBP land cover types

IGBP classification contains the following land cover types:

- **1**: Evergreen needleleaf forests

- **2**: Evergreen broadleaf forests

- **3**: Deciduous needleleaf forests

- **4**: Deciduous broadleaf forests

- **5**: Mixed forests


- **6**: Closed shrublands

- **7**: Open shrublands

- **8**: Woody savannas

- **9**: Savannas

- **10**: Grasslands


- **11**: Permanent wetlands

- **12**: Croplands

- **13**: Urban and built-up

- **14**: Cropland/natural

- **15**: Snow and ice


- **16**: Barren

- **17**: Water bodies

## A4 Table of all input variables

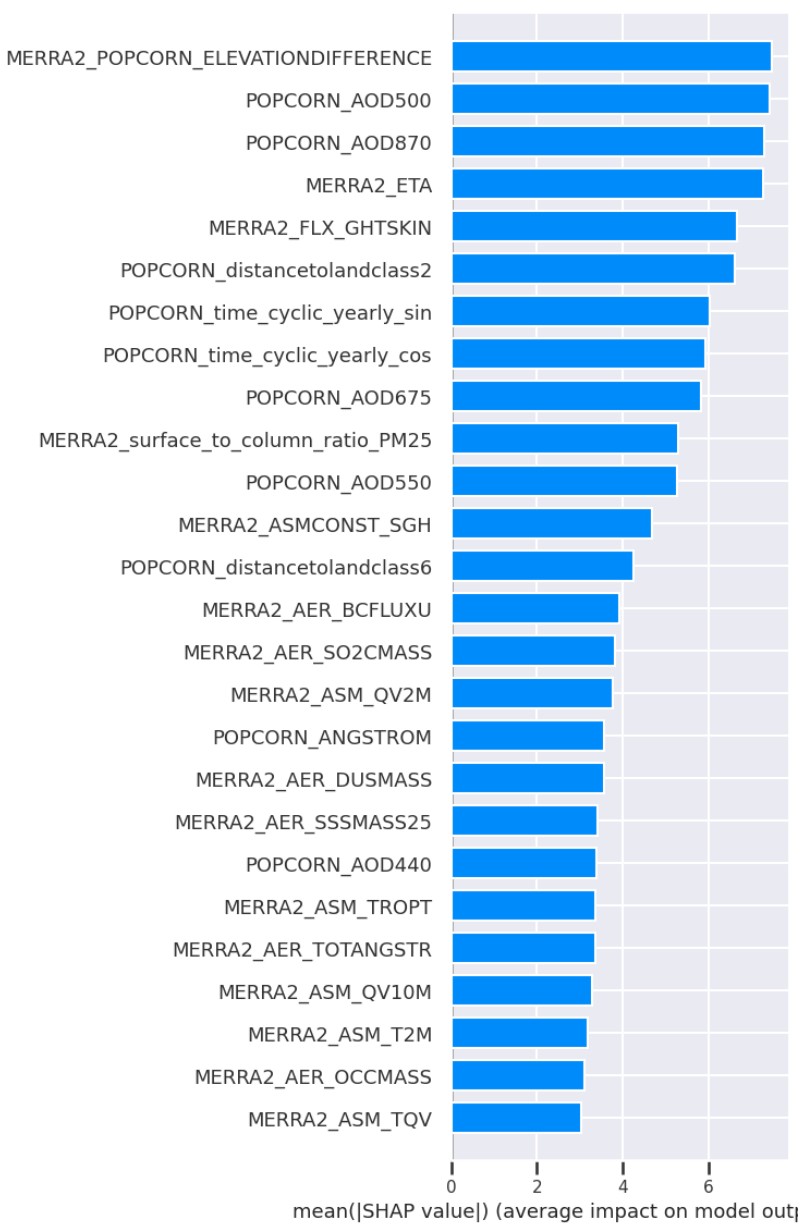

**Figure A1.** Bar plot of the SHAP values for the first 26 input variables in order of importance.

| | | |
|---|---|---|
| MERRA2_POPCORN_ELEVATIONDIFFERENCE | POPCORN_AOD500 | POPCORN_AOD870 |
| MERRA2_ETA | MERRA2_FLX_GHTSKIN | POPCORN_distancetolandclass2 |
| POPCORN_time_cyclic_yearly_sin | POPCORN_time_cyclic_yearly_cos | POPCORN_AOD675 |
| MERRA2_surface_to_column_ratio_PM25 | POPCORN_AOD550 | MERRA2_ASMCONST_SGH |
| POPCORN_distancetolandclass6 | MERRA2_AER_BCFLUXU | MERRA2_AER_SO2CMASS |
| MERRA2_ASM_QV2M | POPCORN_ANGSTROM | MERRA2_AER_DUSMASS |
| MERRA2_AER_SSSMASS25 | POPCORN_AOD440 | MERRA2_ASM_TROPT |
| MERRA2_AER_TOTANGSTR | MERRA2_ASM_QV10M | MERRA2_ASM_T2M |
| MERRA2_AER_OCCMASS | MERRA2_ASM_TQV | MERRA2_FLX_QLML |
| MERRA2_AER_SUFLUXV | MERRA2_FLX_USTAR | MERRA2_AER_SO4CMASS |
| POPCORN_distancetolandclass17 | MERRA2_AER_DUCMASS | MERRA2_AER_BCSMASS |
| MERRA2_AER_BCSCATAU | MERRA2_AER_DUEXTTAU | MERRA2_FLX_EFLUX |
| MERRA2_AER_SO4SMASS | MERRA2_FLX_EVAP | MERRA2_FLX_NIRDR |
| MERRA2_FLX_HFLUX | POPCORN_ASTERDEM | MERRA2_AER_SUANGSTR |
| MERRA2_ASM_TROPPB | MERRA2_AER_BCFLUXV | MERRA2_FLX_TLML |
| MERRA2_FLX_QSTAR | POPCORN_time_cyclic_daily_sin | MERRA2_AER_DUSCATAU |
| MERRA2_FLX_PBLH | POPCORN_distancetolandclass7 | POPCORN_distancetolandclass12 |
| MERRA2_AER_OCSCATAU | MERRA2_AER_TOTEXTTAU | POPCORN_distancetolandclass15 |
| MERRA2_ASM_TROPPV | MERRA2_SURFACERH | MERRA2_FLX_RHOA |
| MERRA2_AER_BCEXTTAU | MERRA2_FLX_FRCLS | MERRA2_AER_DUEXTT25 |
| MERRA2_ASM_T10M | MERRA2_ASM_TS | MERRA2_FLX_SPEED |
| MERRA2_AER_BCANGSTR | MERRA2_AER_DUSCAT25 | MERRA2_AER_OCFLUXU |
| MERRA2_CTMCONST_FRLANDICE | MERRA2_AER_DUCMASS25 | MERRA2_AER_OCEXTTAU |
| MERRA2_FLX_FRCAN | MERRA2_ASMCONST_FRLAND | MERRA2_AER_SSCMASS |
| MERRA2_AER_TOTSCATAU | MERRA2_AER_BCCMASS | MERRA2_CTMCONST_FRACI |
| MERRA2_AER_DUSMASS25 | POPCORN_distancetolandclass16 | POPCORN_CALIOP_MASK_AOD_90_Percent_Below |
| POPCORN_time_cyclic_daily_cos | POPCORN_distancetolandclass4 | MERRA2_AER_DUANGSTR |
| MERRA2_FLX_SPEEDMAX | MERRA2_CTMCONST_FRLAND | MERRA2_FLX_HLML |
| MERRA2_AER_DUFLUXV | MERRA2_AER_OCANGSTR | MERRA2_FLX_TAUY |
| MERRA2_FLX_FRCCN | MERRA2_PM25 | MERRA2_ASMCONST_FRLAKE |
| POPCORN_distancetolandclass8 | MERRA2_AER_SSFLUXV | MERRA2_AER_SUFLUXU |
| MERRA2_FLX_CDQ | POPCORN_distancetolandclass13 | MERRA2_FLX_TSTAR |
| MERRA2_FLX_CN | MERRA2_ASM_V50M | MERRA2_AER_SSSCATAU |
| MERRA2_FLX_QSH | MERRA2_FLX_Z0H | MERRA2_ASM_PS |
| MERRA2_AER_SSEXTTAU | MERRA2_FLX_TCZPBL | MERRA2_AER_OCSMASS |
| MERRA2_FLX_TSH | POPCORN_distancetolandclass3 | MERRA2_SURFACEELEVATION |
| MERRA2_ASM_TROPQ | MERRA2_FLX_CDH | MERRA2_FLX_PGENTOT |
| MERRA2_ASM_U10M | MERRA2_FLX_ULML | MERRA2_ASM_TOX |
| MERRA2_AER_DMSCMASS | POPCORN_distancetolandclass1 | POPCORN_distancetolandclass14 |
| MERRA2_FLX_TAUX | MERRA2_ASMCONST_FRLANDICE | MERRA2_AER_SUSCATAU |
| MERRA2_AER_DUFLUXU | POPCORN_distancetolandclass10 | MERRA2_FLX_PREVTOT |
| MERRA2_CTMCONST_FROCEAN | MERRA2_ASM_TQL | MERRA2_ASM_U2M |
| MERRA2_ASM_DISPH | MERRA2_FLX_PRECTOT | MERRA2_AER_SO2SMASS |
| MERRA2_FLX_CDM | MERRA2_FLX_Z0M | MERRA2_ASM_windspeed |
| POPCORN_distancetolandclass11 | MERRA2_FLX_DISPH | MERRA2_AER_OCFLUXV |
| MERRA2_FLX_PRECTOTCORR | MERRA2_ASM_TROPPT | MERRA2_FLX_PRECLSC |
| MERRA2_FLX_BSTAR | MERRA2_ASM_TO3 | POPCORN_CALIOP_MASK_AOD_63_Percent_Below |
| MERRA2_FLX_PRECCON | MERRA2_ASM_TQI | MERRA2_ASMCONST_FROCEAN |
| MERRA2_CTMCONST_PHIS | POPCORN_distancetolandclass5 | MERRA2_CTMCONST_FRLAKE |
| MERRA2_FLX_TAUGWX | MERRA2_FLX_PRECANV | MERRA2_ASM_V2M |
| MERRA2_ASMCONST_PHIS | MERRA2_FLX_NIRDF | POPCORN_distancetolandclass9 |
| MERRA2_ASM_SLP | POPCORN_BlackMarble | POPCORN_distancetoroad_upwind |
| MERRA2_AER_SSANGSTR | MERRA2_FLX_VLML | MERRA2_AER_SSSCAT25 |
| MERRA2_ASM_winddirection | MERRA2_FLX_TAUGWY | MERRA2_AER_SSFLUXU |
| MERRA2_AER_SUEXTTAU | MERRA2_ASM_V10M | MERRA2_AER_SSCMASS25 |
| MERRA2_FLX_PRECSNO | MERRA2_AER_SSEXTT25 | MERRA2_AER_DMSSMASS |
| MERRA2_FLX_RISFC | MERRA2_AER_SSSMASS | MERRA2_ASM_U50M |
| MERRA2_FLX_FRSEAICE | | |

**Table A1.** List of input variables used in our model ordered by SHAP value (from left to right and from top to bottom).

## A5   Comparison post-process correction approach vs fully-learned approach

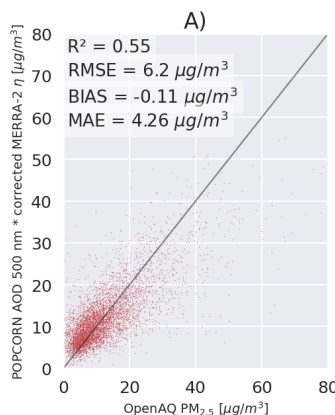 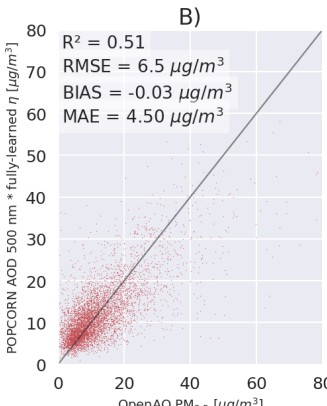

**Figure A2.** A) Post process corrected PM$_{2.5}$ predictions against OpenAQ PM$_{2.5}$ measurements. B) Fully-learned NOODLESALAD PM$_{2.5}$ predictions against OpenAQ PM$_{2.5}$ measurements.

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
