# Peer review of "Post-process correction improves the accuracy of satellite $\ensuremath{\text{PM}_{2.5}}$ retrievals"

_EGUsphere, 2023_

## Author Comment (AC1)

**Post-process correction improves the accuracy of satellite PM$_{2.5}$ retrievals manuscript - Reply to Adam Povey, Supriya Mantri and Laura Horton**

Andrea Porcheddu, Ville Kolehmainen, Timo Lähivaara, Antti Lipponen

We would like to thank you for reading the manuscript and giving comments. Below we answer to each of the comments.

**1 Answers**

**1.1 The introduction was clear with good motivations and references but, in line 25, the reference to the WHO air quality guideline does not specify the time frame. Is it an annual average?**

5 Yes, it is an annual average.

**1.2 Is it possible to give any additional reasoning behind the use of hourly downscaling of daily PM averages? This strikes us as one of the more consequential choices in this methodology and neglects complexities such as the diurnal cycle. (We aren't questioning the authors' judgement, nor disagreeing; merely curious as we wouldn't have thought of this.)**

10 We are simply unrolling 24h averages given every hour, so the hourly measurements information is contained in each 24h average.

**1.3 In section 2.3 the authors mention they use MERRA-2 reanalysis variables as an input for the model and provide a lengthy list in the appendix. Have all of the available variables been used as an input or are there some variables not included? It may be helpful for those designing similar algorithms if you also mention which variables were not included and, if so, why not?**

15

A table with all the features used in the training of the model is available at the end of the manuscript. We have used all the MERRA-2 reanalysis variables mentioned in the table. We have also carried out SHAP analysis that showed us that all MERRA-2 input variables were informative and had non-negligible effect on the model output.

**1.4 Could you provide a reference for the RH equation on line 93? Or is it at standard temperature/pressure?**

20 The RH equation is based on the Clausius-Clapeyron equation. We have now mentioned the Clausius-Clapeyron equation and included a reference to a paper using the formula and having more information about the equation in the revised manuscript.

**1.5 In the NASA Black Marble Night Light section (2.6.2), the authors could include a reference to the data set used, especially a DOI so we can distinguish which of the four available datasets was used.**

We have added a reference to the dataset in the revised manuscript.

**1.6 We found Sections 3.3 and 3.4 quite opaque. They could be improved by adding more specific details about the process, such as the equations used as an input to the correction model. A flow chart of the steps used in the approach would make it more clear for an audience which is less familiar with machine learning and neural networks.**

The sections have been improved adding some more equations and explanations of their content.

**1.7 Those of us with little experience with neural networks did not understand what Figure 2 wished to convey and those of us used to neural networks felt Figure 2 adds little to the text; it could be removed.**

Thank you for the suggestion but we decided to keep the figure since we think it could be informative.

**1.8 At line 183, it would be helpful to have an understanding of what is meant by "slightly better"?**

We modified the text from "slightly better" to "better".

**1.9 Figure 3 could be improved as it is hard to distinguish between lines, perhaps using a filled histogram of stacked bars.**

We modified the plot with thicker lines and no transparency for the colors.

**1.10 We are curious if the authors considered any methods to amplify the availability of high PM2.5 observations, such as data augmentation? Our understanding was that the balancing of training data is an important step in constructing a neural network to recognise rare events and we would value the authors' opinion.**

We have tested augmenting the data using CTGAN (Conditional Tabular Generative Adversarial Network) and TVAE (Tabular Variational Auto Encoder) (Xu et al., 2019) but didn't observe any improvement.

**1.11 From line 223, what do you mean by "the fully learned approach were less accurate than with the post correction approach"? What metric of accuracy was used and how significant was the difference? This would help guide our own efforts in neural network generation.**

We calculated different metrics including $R^2$, RMSE and MAE. In particular, for the post-correction approach we obtained $R^2$=0.55, RMSE=6.2 $\mu g/m^3$ and MAE=4.26 $\mu g/m^3$, while for the fully-learned approach we obtained $R^2$=0.51, RMSE=6.5 $\mu g/m^3$ and MAE=4.50 $\mu g/m^3$.

**1.12    In Figure 4, are the authors certain about the plotting of panel B? Compared to panel A there appears to be some duplication of points up the y-axis. For example, there are three points at the extreme right of (A) but over ten in (B). This effect is not exhibited in panels D-F.**

Yes, we are sure about the plotting. We checked further and we found no problems.

**1.13    In Figure 5 (comparison of the uncorrected and corrected methods at the ground stations), do the authors have any understanding of why there is a large discrepancy between OpenAQ and both satellite estimates for most of the sites (i.e. the blue dots do not overlap the red line in 5/9 cases shown). Is it because of some local source (e.g. roads or small industrial buildings) in close proximity of the stations that isn't present in Madrid?**

We looked at the correlations between input features and errors and we did't find any single feature that could explain the errors for the shown satellite overpass. Please note that the monthly averages for the test station in Paris are well aligned with the ground based data.

**1.14    In Figures 5 and 6, could the dots representing sites and arrows indicating one be made substantially larger and outlined with a colour not in the plot (such as green or blue)? Our older member had failed to see them on his own.**

The figures have been modified now, we chose green colour and larger arrow/font.

**1.15    In Figure 7 bottom left (Hull Freetown) why is the uncertainty so large April? What are the possible reasons for high PM2.5 in February?**

We don't have any explanation for the large uncertainty in April: we looked at correlations between errors and input data but we didn't find anything interesting. For what regards the high $PM_{2.5}$ in February, it could be caused by the lower boundary layer height in winter months (as other causes like house heating, etc.).

**1.16    If practical, it would be interesting to add an appendix highlighting a few ensemble members for Fig 5 and/or 6. This would demonstrate if the smoothness of the fields shown at the top right of those figures is due to the action of the neural network or due to the median filter over the ensemble.**

The smoothness of the fields is not due to the median filter since the same smoothness can be seen in the figure regarding the not-corrected method (where no median filter is applied).

**1.17    It would be interesting to hear if training the model over an area with higher PM2.5 levels, such as South Africa or India, and then testing over central Europe improves the model's performance, particularly for higher PM2.5 values.**

That's our plan for future studies, we are curious to see if gathering more data from all around the world could improve the modelling in Europe.

**References**

Xu, L., Skoularidou, M., Cuesta-Infante, A., and Veeramachaneni, K.: Modeling Tabular data using Conditional GAN, 2019.

---

## Author Comment (AC2)

**Post-process correction improves the accuracy of satellite PM$_{2.5}$ retrievals - Reply to referees**

Andrea Porcheddu, Ville Kolehmainen, Timo Lähivaara, Antti Lipponen

We would like to thank the reviewers for reading carefully the manuscript and giving their comments. Below we reply to each of the comments.

**1 Answers to reviewer #1**

**1.1 There are many researches that focus on using AOD to estimate PM$_{2.5}$ through machine learning approaches.**
5     **Compared with them, what is the innovation of this study? I understand that this study corrects the ratio between PM$_{2.5}$ and AOD derived from MERRA2 and applies the improved ratio to satellite AOD to estimate surface PM$_{2.5}$ concentrations, which is different from other researches that estimate PM$_{2.5}$ directly. Although this is a new approach, what is the advantage of this study. Compared with previous researches, can the new approach provide better PM$_{2.5}$ estimation?**

10   The novelty of this study is to employ the post process correction where we use machine learning for correcting the geophysical model based AOD-to-PM$_{2.5}$ conversion ratio instead of directly predicting the conversion ratio. The rationale for this selection is improved accuracy over the conventional approach of learning the conversion ratio directly. Figure 1 shows the results obtained with the conventional approach of learning directly the AOD-to-PM$_{2.5}$ ratio. The $R^2$, RMSE and MAE with the proposed post-correction approach (left image) are better than with the conventional direct estimation (right image).

15   **1.2 This exclude the PM$_{2.5}$ concentrations that are larger than 80 $\mu g/m^3$. This would reduce the importance of this study as the research community is more interested in heavy polluted scene. The authors excluded that condition that PM$_{2.5}$ concentrations that are larger than 80 $\mu g/m^3$ due to imbalanced data (only a small set of data with PM$_{2.5}$ concentrations that are larger than 80 $\mu g/m^3$). Can the problem is solved through bagging or other approaches?**

20   The PM$_{2.5}$ values beyond 80 $\mu g/m^3$ were excluded from the study due to sparsity of high value data in the region of interest (central Europe, 2019) considered in this study. Similar cutoff for high PM$_{2.5}$ values has been applied, for example, in Ibrahim et al. (2022)). To address the lack of high PM$_{2.5}$ data, one could think of producing synthetic training data using machine learning models such as TVAE (Tabular Variational Auto Encoder) or CTGAN (Conditional Tabular Generative Adversarial Network) (Xu et al., 2019): we tried already to use them to balance the data but they did not improve the results. In future studies

[Figure]

[Figure]

**Figure 1.** A) Post process corrected PM$_{2.5}$ predictions against OpenAQ PM$_{2.5}$ measurements. B) Fully-learned NOODLESALAD PM$_{2.5}$ predictions against OpenAQ PM$_{2.5}$ measurements.

25  the approach could be extended to more global training data and include data, for example, from India and China, where higher values of PM$_{2.5}$ exist more frequently.

**1.3    More information of satellite data is needed. What is the temporal resolution and swath of the sensor?**

Two Sentinel-3 satellites currently flying provide revisit times of less than two days for OLCI and less than one day for the SLSTR instrument at equator. Swath width of the OLCI instrument is 1270 km. SLSTR swath width is 1420 km for the nadir
30  view and 750 km for the oblique view. In our study, we base our aerosol information on the official Sentinel-3 Synergy Land data product and the characteristics of that data set matches our satellite overpass data. The information have been added to the manuscript.

**1.4    This study only demonstrates the validations in Fig. 4. It would be more interesting to show the fitting (training) as well.**

35  Fig.2 shows the results on the training set. The metrics are better than on the test set (as it is expected to be), but not very different since we used the early stopping technique as regularization to avoid overfitting on the training set. We remark that in this study we separated part of the available stations to be used as independent validation data for the methods. For operational use, the model could be trained using all the available data in the training and test data sets.

**1.5    In Fig.4, why monthly mean shows larger bias than instant estimations?**

40  It is not strictly necessary that the monthly bias is smaller than the instantaneous bias. We have added mean absolute error (MAE) as an additional metric in the figures. MAE shows improvement for the monthly data over the instantaneous data.

[Figure]

**Figure 2.** A) MERRA-2 PM$_{2.5}$ predictions against OpenAQ PM$_{2.5}$ measurements per single-overpass. B) Uncorrected NOODLESALAD PM$_{2.5}$ predictions against OpenAQ PM$_{2.5}$ measurements per single-overpass. C) Corrected NOODLESALAD PM$_{2.5}$ predictions against OpenAQ PM$_{2.5}$ measurements per single-overpass. These results regard the training set.

**1.6 Latitude and longitude are missing in the top panels of Figs. 5 and 6.**

Latitude and longitude have been added to the panels.

**1.7 Line 78-80: I cannot understand. More details are needed for the method description.**

45 Some OpenAQ stations report 24 hour average PM$_{2.5}$ every hour based on the last 24 hours.

In this work, we used the 24 hour averages given every hour to estimate hourly PM$_{2.5}$. This processing was done station-by-station using a Tikhonov regularized (with regularization parameter value 0.05) least-squares fit to unfold the time integrated data into hourly estimates.

In practice, the hourly PM$_{2.5}$ estimates were computed using the formula

$$\quad PM_{2.5,1h} = \left(A^T A + \alpha I\right)^{-1} A^T b, \tag{1}$$

where the system matrix

$$
A = \begin{bmatrix}
\frac{1}{24} & \frac{1}{24} & \cdots & \frac{1}{24} & 0 & 0 & \cdots & 0 \\
0 & \frac{1}{24} & \cdots & \frac{1}{24} & \frac{1}{24} & 0 & \cdots & 0 \\
& & & \vdots & & & & \\
0 & 0 & \cdots & 0 & 0 & 0 & \cdots & \frac{1}{24}
\end{bmatrix}, \tag{2}
$$

is 24 hour time averaging operator and the data vector

$$
b =
\begin{bmatrix}
PM_{2.5,24h,1} \\
PM_{2.5,24h,2} \\
\vdots \\
PM_{2.5,24h,N}
\end{bmatrix},
\tag{3}
$$

55

$$
PM_{2.5,1h} =
\begin{bmatrix}
PM_{2.5,1h,24} \\
PM_{2.5,1h,25} \\
\vdots \\
PM_{2.5,1h,N}
\end{bmatrix}
\tag{4}
$$

contain the hourly 24 hour averages $PM_{2.5,24h,N}$ of the station data and $\alpha = 0.05$ is the regularization parameter. The solution vector $PM_{2.5,1h,N}$ contains the unfolded 1 hour PM$_{2.5}$ at timestep $N$, respectively.

We have added this explanation in the revised manuscript.

60 **1.8   Why CALIOP data are used. This is monthly mean data, but PM$_{2.5}$ and AOD has strong diurnal variation. Can the CALIOP data help to improve PM$_{2.5}$ estimation?**

We tested the approach without CALIOP data in the training and the result is shown in fig.3 (left image: CALIOP included, right image: without CALIOP). The model which uses CALIOP data results in slightly better accuracy, indicating that use of CALIOP data is warranted.

[Figure]

[Figure]

**Figure 3.** A) Corrected NOODLESALAD PM$_{2.5}$ predictions against OpenAQ PM$_{2.5}$ measurements per single-overpass (using CALIOP data). B) Corrected NOODLESALAD PM$_{2.5}$ predictions against OpenAQ PM$_{2.5}$ measurements per single-overpass (without CALIOP data).

65 **1.9 The unit of RMSE is missing throughout the paper.**

The unit of RMSE has been added to the paper.

**2 Answers to reviewer #2**

**2.1 Abstract: The study lacks major conclusions and quantitative descriptive results.**

The abstract has been modified as follows (extension highlighted in red):

70     Estimates of $PM_{2.5}$ levels are crucial for monitoring air quality and studying the epidemiological impact of air quality on the population. Currently, the most precise measurements of $PM_{2.5}$ are obtained from ground stations, resulting in limited spatial coverage. In this study, we consider satellite-based $PM_{2.5}$ retrieval, which involves conversion of high-resolution satellite retrieval of Aerosol Optical Depth (AOD) into high-resolution $PM_{2.5}$ retrieval. To improve the accuracy of the AOD to $PM_{2.5}$ conversion, we employ the machine learning based post-process correction to correct the AOD-to-PM conversion ratio derived

75 from Modern-Era Retrospective analysis for Research and Applications, Version 2 (MERRA-2) reanalysis model data. The post-process correction approach utilizes a fusion and downscaling of satellite observation and retrieval data, MERRA-2 reanalysis data, various high resolution geographical indicators, meteorological data and ground station observations for learning a predictor for the approximation error in the AOD to $PM_{2.5}$ conversion ratio. The corrected conversion ratio is then applied to estimate $PM_{2.5}$ levels given the high-resolution satellite AOD retrieval data derived from Sentinel-3 observations. The region of

80 study is central Europe during the year 2019. Our model produces $PM_{2.5}$ estimates with a spatial resolution of 100 meters at satellite overpass times with $R^2 = 0.55$ and RMSE = 6.2 $\mu g/m^3$. The corresponding metrics for monthly averages are $R^2 = 0.72$ and RMSE = 3.7 $\mu g/m^3$. Additionally, we have incorporated an ensemble of neural networks to provide error envelopes for machine learning related uncertainty in the $PM_{2.5}$ estimates. The proposed approach can produce accurate high resolution $PM_{2.5}$ data that can be very useful for air quality monitoring, emission regulation and epidemiological studies.

85 **2.2 The introduction is very short and lacks a comprehensive review of numerous previous studies on converting AOD to $PM_{2.5}$ using machine learning models.**

The introduction has been modified as follows (extension highlighted in red):

[revised manuscript text omitted]

**2.3 The use of MERRA2-2 for calculating PM$_{2.5}$ is criticized for its inaccuracies and omission of certain species like Nitrate. It is suggested to consider using GEOS-CF data, which provides PM$_{2.5}$ simulations at a higher resolution of 0.25 degrees.**

We thank the referee for the suggestion. We agree that GEOS-CF would be a suitable and good model data to consider in our methodology. Our methodology developed is not restricted to any single model or satellite data. Some criteria in selecting the model data for our work were long time series and widely used model in scientific literature and we therefore ended up selecting MERRA-2. In our future work, we will consider using GEOS-CF data as it has somewhat better spatial accuracy and more relevant species for air quality applications.

**2.4 The spatial resolution of high-resolution indicators such as roads and nighttime lights needs clarification.**

– NASA Black Marble Night Lights: We use the 500 meter resolution data.

– OpenStreetMap roads: The original data is vector data with a typical accuracy of orders of meters. We have re-projected the OpenStreetMap road data to 100 meter resolution before use.

We have added the resolutions used to the revised manuscript.

**2.5 The excessive number of variables selected raises questions about their relevance and contribution to the network model. It would be beneficial to employ importance analysis methods to identify and eliminate redundant variables. This process will streamline the model and improve its efficiency and interpretability.**

We used SHAP analysis (Lundberg and Lee (2017)) to estimate the feature importance after model training. A bar plot can be seen in fig.4 for the first 26 features found with the SHAP analysis. Table 1 contains all the input variables listed by importance (SHAP value). Since all the variables had non-negligible SHAP values, indicating some information content in them, we decided to keep them all. The input variable table in the manuscipt has been modified so that now the variables are listed in order of importance by the SHAP values.

[Figure]

**Figure 4.** Bar plot of the SHAP values for the first 26 input variables in order of importance.

**2.6 Figure 3: It is unclear how the training and validation stations are divided. Additionally, the proximity of stations may lead to correlation issues, affecting the independence of training and testing samples spatially.**

We are not the first ones to use site-validation, please see the review of different validation methods used in the literature Tang et al. (2024). Essentially we divided randomly the stations into training set, validation set and test set, then used the related data

165 accordingly. For what regards the correlation issues, there should be only a minimal effect since we are operating at resolution of 100 m and there are not two or more stations in the same pixel.

**2.7 Section 3.4: The rationale for choosing the neural network model over other more powerful machine learning and deep learning models is not provided. The advantages of this model should be discussed.**

Using a fully connected neural network compared to other common models in $PM_{2.5}$ prediction like random forest is very
170 suitable in the case one is working with assumption of independent pixels and a high number of data samples and features. In this case, our study is limited to the chosen ROI and period of time (2019) so we have roughly 20000 points in the training set: our plan is to extend the ROI and the period of time in future studies so to have more data in the training set, where the fully connected neural network should show all its learning capability. Comparing the fully connected neural network to other deep learning models, we think it is the most suitable architecture for the task at hand since we are dealing with tabular data and a fair
175 amount of features. One could for example think of using a convolutional neural network and reorganizing the data samples into appropriate matrices but still the number of features is small and we don't know if we would benefit deploying a convolutional network (there's no computational burden to justify this approach and the fully connected neural network should be able to find proper representations of the input data in its hidden layers).

**2.8 Figure 4: While the accuracy has improved, the correlation remains relatively low (only 0.63), compared to**
180 previous studies achieving higher accuracy with AI (R2 higher than 0.8). The significance of the study is questioned, and comparison with previous studies to assess improvement is recommended.

The $R^2$ coefficient is low compared to other studies but we should compare our manuscript to papers who deal with, for example. the same ROI, spatial resolution and study time period. Our RMSE is comparable to other studies that consider European countries or similar ROI (Schneider et al. (2020); Ibrahim et al. (2022); Handschuh et al. (2023)). The $R^2$ coefficient is low
185 but the number of data samples at hand is low too: as mentioned before, we are considering the year 2019 and compared, for example, to Ibrahim et al. (2022) we have $10\%$ of data. Our ensemble of fully connected neural networks would benefit from having more data and the metrics would improve further. Also the data and the preprocessing choices are different, so the studies are not directly comparable.

[revised manuscript text omitted]

---

## Author Response (AR2)

**Post-process correction improves the accuracy of satellite PM$_{2.5}$ retrievals - Authors response**

Andrea Porcheddu, Ville Kolehmainen, Timo Lähivaara, Antti Lipponen

We would like to thank the reviewers for reading carefully the manuscript and giving their comments. Here below we report the changes we have made to the manuscript. In the following pages the questions from the reviewers and the related answers from the authors can be found. At the end of the document we report a version of the manuscript realized with latexdiff in order to show explicitly the changes.

**1 Changes made following reviewer #1 questions**

- We added a comment about why the post-process correction approach should lead to improvement compared to a fully-learned approach at the end of Section 3.2.

- We added figure A2 showing the comparison of the proposed approach vs. fully learned approach.

- We updated the manuscript writing explicitly that the WHO guideline is an annual average (line 27).

**2 Changes made following reviewer #2 questions**

- At the end of section 4, we added a comment about the use of GEOS-CF data, the result obtained (shown in figure 8) and the comparison with what we previously achieved with MERRA-2 data.

**Post-process correction improves the accuracy of satellite PM$_{2.5}$ retrievals - Reply to referees**

Andrea Porcheddu, Ville Kolehmainen, Timo Lähivaara, Antti Lipponen

We would like to thank the reviewers for reading carefully the manuscript and giving their comments. Below we reply to each of the comments.

**1 Answers to reviewer #1**

**1.1 Although the authors have shown the new approach is a little better than the approaches that using direct AOD-PM2.5 relationship through scatter plots in the reply, some discussion about why the new approach performs better is needed.**

Our model estimates the approximation error of a physics based retrieval algorithm, so it doesn't estimate directly the mapping from the measurement data to the retrieval quantity as a fully-learned approach would do. The reason why the proposed approach performs better comes from the approximation error of the physics based retrieval being a less complicated function (than the direct retrieval) for a machine learning regression to learn. This leads to a more accurate and reliable estimation of the retrieval quantity. We have added a comment about this at the end of Section 3.2, and we have also added the comparison of the proposed approach vs. fully learned approach as figure A2 in the manuscript.

**1.2 Although the authors have replied all the comments, it's better to make corresponding revisions. The figures in the response can be added to the manuscript or supplements. The authors clarified that "it is an annual average" in the reply on CC1, but corresponding revision was not added in the manuscript. These are just some examples; authors may consider making corresponding revisions for the replies.**

Referring to our reply to your previous comment, the figure showing the comparison between post-process correction approach and fully-learned approach has been added to the manuscript. We updated the manuscript writing explicitly that the WHO guideline is an annual average.

**2   Answers to reviewer #2**

**2.1   The authors did not address my main comments, especially regarding the updated data using GEOS-CF/FP data. As the current study focuses on central Europe with 100 m? during the year 2019 rather than historical long time series, GEOS-CF data offers more additional species like Nitrate and ammonium, along with other factors (e.g., AOD). GEOS-CF/FP data should be used instead, which can provide PM2.5 and other simulations at a much higher resolution than MERRA2.**

As suggested, we tested the performance of the post-process correction with GEOS-CF model data instead of MERRA-2 data. Using the GEOS-CF data, We trained an ensemble of 80 fully-connected feed-forward neural networks. The result is shown in fig.1, and it has been added as Figure 8 in the revised manuscript. The post-process correction using GEOS-CF data lead to similar performance as with the MERRA-2 data with MERRA-2 based approach having slightly better error metrics, as evident by comparison of fig.1 and Fig 4C in the manuscript.

[Figure]

**Figure 1.** A) GEOS-CF $PM_{2.5}$ predictions against OpenAQ $PM_{2.5}$ measurements per single-overpass. B) Uncorrected NOODLESALAD $PM_{2.5}$ predictions against OpenAQ $PM_{2.5}$ measurements per single-overpass. C) Corrected NOODLESALAD $PM_{2.5}$ predictions against OpenAQ $PM_{2.5}$ measurements per single-overpass.

**2.2   Feature Importance: The authors did not clarify the physical aspects behind their results, such as why $MERRA2\_POPCORN\_ELEVATIONDIFFERENCE$ is identified as the most important feature.**

Our method is a data-driven approach, and the feature importance metrics are data based estimates that reflect the importance of the data features in the training of the generic (non-physical) feed forward neural network model. In the present case, the neural network model is trained to predict the approximation error in a conventional physics based retrieval algorithm which contains

model simplifications and approximations for the auxiliary model parameters. Due to these reasons, it is not possible to give definite physical reasoning for the effects and possible joint effects of the different data features in the performance of the neural network model for the prediction of the approximation error.

**2.3 Validation: There is still a lack of clarity regarding how the training and validation samples were divided. What are the proportions used for each?**

We divided the ground monitoring stations in three different sets. The training set contains $60\%$ of the stations, the validation set $20\%$ of the stations and the test set the remaining $20\%$.

[revised manuscript text omitted]